

# Assimilation of sea ice thickness derived from CryoSat-2 along-track freeboard measurements into the Met Office's Forecast Ocean Assimilation Model (FOAM)

Emma K. Fiedler[1], Matthew Martin[1], Ed Blockley[1], Davi Mignac[1], Nicolas Fournier[1], Andy Ridout[2], Andrew Shepherd[3], and Rachel Tilling[4,5]

[1]Met Office, Exeter, EX1 3PB, UK
[2]Centre for Polar Observation and Modelling, University College London, London, WC1E 6BT, UK
[3]Centre for Polar Observation and Modelling, University of Leeds, Leeds, LS2 9JT, UK
[4]NASA Goddard Space Flight Center, Greenbelt, MD, USA
[5]Earth System Interdisciplinary Center, University of Maryland, College Park, MD, USA

**Correspondence:** Emma Fiedler (emma.fiedler@metoffice.gov.uk)

**Abstract.** The feasibility of assimilating SIT (sea ice thickness) observations derived from CryoSat-2 along-track measurements of sea ice freeboard is successfully demonstrated using a 3D-Var assimilation scheme, NEMOVAR, within the Met Office's global, coupled ocean-sea ice model, FOAM (Forecast Ocean Assimilation Model). The Arctic freeboard measurements are produced by CPOM (Centre for Polar Observation and Modelling) and are converted to SIT within FOAM using

modelled snow depth. This is the first time along-track observations of SIT have been used in this way, with other centres assimilating gridded and temporally-averaged observations. The assimilation greatly improves the SIT analysis and forecast fields generated by FOAM, particularly in the Canadian Arctic. Arctic-wide observation-minus-background assimilation statistics show improvements of 0.75 m mean difference and 0.41 m RMSD (root-mean-square difference) in the freeze-up period, and 0.46 m mean difference and 0.33 m RMSD in the ice break-up period, for 2015-2017. Validation of the SIT analysis against

independent springtime in situ SIT observations from NASA Operation IceBridge shows improvement in the SIT analysis of 0.61 m mean difference (0.42 m RMSD) compared to a control without SIT assimilation. Similar improvements are seen in the FOAM 5-day SIT forecast. Validation of the SIT assimilation with independent BGEP (Beaufort Gyre Exploration Project) sea ice draft observations does not show an improvement, since the assimilated CryoSat-2 observations compare similarly to the model without assimilation in this region. Comparison with Air-EM (airborne electromagnetic induction) combined mea-

surements of SIT and snow depth shows poorer results for the assimilation compared to the control, which may be evidence of noise in the SIT analysis, sampling error, or uncertainties in the modelled snow depth, the assimilated observations, or the validation observations themselves. The SIT analysis could be improved by upgrading the observation uncertainties used in the assimilation. Despite the lack of CryoSat-2 SIT observations over the summer due to the effect of meltponds on retrievals, it is shown that the model is able to retain improvements to the SIT field throughout the summer months, due to previous SIT

assimilation. This also leads to regional improvements in the July SIC (sea ice concentration) of 5% RMSD in the European sector, due to slower melt of the thicker modelled sea ice.



# 1 Introduction

In recent decades, Arctic sea ice cover has undergone a considerable reduction in both thickness and extent (e.g. Comiso et al., 2008; Kwok et al., 2009; Lindsay and Schweiger, 2015; Stroeve and Notz, 2018; Meredith et al., 2019), which has the

potential to impact weather and climate at lower latitudes (e.g. Koenigk et al., 2016; Screen, 2017), alter the ecosystem and living environment (e.g. Meier et al., 2014), and to change the nature of Arctic shipping by opening up new sea routes (e.g. Smith and Stephenson, 2013; Wei et al., 2020; Zeng et al., 2020). Therefore, predictions of sea ice cover on timescales of days, seasons and beyond are becoming increasingly important.

It has long been understood that the initialisation of NWP (numerical weather prediction) models using data assimilation is

vital for producing accurate forecasts over a range of timescales. In the case of sea ice forecasting, assimilation of SIC (sea ice concentration) observations into coupled ocean-sea ice models is well established and routine at almost all centres producing operational sea ice and ocean forecasts (e.g. Bertino and Lisaeter, 2008; Blockley et al., 2014; Smith et al., 2015; Posey et al., 2015; Lemieux et al., 2016). Timeseries of well-homogenised satellite observations of SIC are available dating back to 1979 (e.g. Lavergne et al., 2019) allowing for accurate monitoring of SIC and extent, and use in hindcasts and reanalyses.

Large-scale observations of SIT (sea ice thickness) have become available much more recently than SIC, with the first dedicated polar satellite mission, ESA's CryoSat-2, launched in April 2010. CryoSat-2 is able to observe sea ice freeboard (the height of sea ice above the ocean surface) and, based on the assumption that the sea ice is floating in hydrostatic equilibrium, the freeboard observations can be converted to SIT. Although observations of sea ice freeboard have been successfully obtained from previous radar and laser altimetry missions, namely ESA's ERS (European Remote Sensing) and Envisat satellites, and

NASA's ICESat (Ice, Cloud and land Elevation Satellite) (e.g. Laxon et al., 2003; Kwok and Rothrock, 2009), there are considerably larger unobserved areas over the poles than for CryoSat-2. Observations from ICESat were limited by cloud cover and, for the radar missions, instrument footprint sizes are notably larger than for CryoSat-2, with a greater contribution of noise in the observations from radar speckle (Laxon et al., 2013). The only additional estimates of SIT are sparse in situ observations from surface and submarine platforms. Consequently, the processing of satellite freeboard observations is still very much an

ongoing and active area of research and development, particularly regarding the accuracy of observations, the characterisation of uncertainties and the timeliness of data delivery (e.g. Ricker et al., 2014, 2016; Tilling et al., 2016, 2018). The field is additionally much less mature than the use of ocean altimetry to measure sea level anomaly, which is also readily assimilated into operational ocean forecasting models (e.g. Blockley et al., 2014).

As a complement to radar altimetry, which can be used to retrieve SIT greater than around 1 m, observations of thin SIT

can be obtained from the SMOS (Soil Moisture and Ocean Salinity) satellite, launched in November 2009. Using a passive microwave radiometer, the thickness of sea ice under 1 m thick is inferred from retrievals of brightness temperature (Kaleschke et al., 2010; Tian-Kunze et al., 2014).

A number of studies have emphasised the importance of accurate initialisation of SIT fields for seasonal predictions of sea ice concentration and extent: Day et al. (2014); Massonnet et al. (2015); Collow et al. (2015); Dirkson et al. (2017); and CryoSat-

2 and/or SMOS SIT observations have been used to initialise seasonal sea ice forecasts by Blockley and Peterson (2018);





Yang et al. (2019); Allard et al. (2020). Several studies have demonstrated the impact of using satellite SIT observations in addition to SIC to initialise short-term operational sea ice forecasts: e.g. Yang et al. (2014); Mu et al. (2018); Xie et al. (2018); Liang et al. (2020). All of these previous studies have made use of gridded, temporally-averaged satellite SIT datasets (e.g. weekly, monthly) and not the along-track CryoSat-2 data used herein. It was not originally envisioned that it would be possible to use CryoSat-2 observations without a certain degree of spatial and temporal averaging, due to noise in the freeboard retrievals (Wingham et al., 2006). However, the nature of operational ocean forecasting at the Met Office, and developments towards a coupled ocean-ice-atmosphere NWP framework with short assimilation time windows of the order of 6 hours, means that along-track rather than temporally-averaged observations are required. Additionally, the use of observations without gridding or temporal averaging aims to improve analysis quality by reducing spatially-correlated uncertainties in the measurements, and allowing uncertainty estimates, vital for data assimilation, to be more easily determined than would be for data with more processing applied.

Therefore, in this study we investigate the feasibility of assimilating Arctic SIT observations derived from along-track CryoSat-2 radar altimeter sea ice freeboard measurements into a global, coupled ocean-sea ice model. We demonstrate that the assimilation system (including prior observation quality control) successfully reduces the effect of noise in the observations such that initial gridding and temporal averaging are not required. Work is currently underway at the Met Office to assimilate SMOS observations in conjunction with those from CryoSat-2, and will be reported in a future publication. A validation of the SIT and SIC analyses, 1-day and 5-day forecasts generated using the FOAM system assimilating CryoSat-2 SIT, in addition to all the standard observation types, is presented. Improvements in analysis and forecast performance compared to a control without SIT assimilation are demonstrated. The paper is structured as follows: Sect. 2 includes descriptions of the FOAM system, observations used in this study, and assimilation methods. Results from the SIT assimilation experiment are shown in Sect. 3, and validation using assimilation statistics and independent in situ observations follow in Sects. 4 and 5 respectively. A final discussion and conclusions are presented in Sect. 6.

## 2 Methods

### 2.1 The FOAM test system

FOAM (Forecast Ocean Assimilation Model; Blockley et al., 2014) is the Met Office's global, coupled ocean-sea ice model. It is forced at the surface using output from the Met Office NWP system. The ocean and sea ice components of FOAM are also used in an ocean-sea ice-atmosphere-land coupled short-range forecasting system (Guiavarc'h et al., 2019). Analyses and 5-day forecasts of ocean and sea ice variables are produced operationally from the coupled system, and are disseminated through CMEMS (Copernicus Marine Environment Monitoring Service; marine.copernicus.eu). FOAM analyses are also used operationally to initialise the Met Office's seasonal forecasting system, GloSea (MacLachlan et al., 2014). Here we focus on the forced ocean and sea ice FOAM system, but the implementation of any developments will also be of benefit to the coupled short-range and seasonal prediction systems.



The ocean model component of FOAM, NEMO (Nucleus for European Modelling of the Ocean; Madec, 2017) is coupled to the Los Alamos Sea Ice Model, CICE (Hunke et al., 2015). The FOAM system has recently been upgraded to 1/12 degree resolution for both ocean and sea ice components (ORCA12; Barbosa Aguiar et al., 2021), but the version of FOAM used herein has a 1/4 degree tripolar grid (ORCA025), with 75 vertical levels in the ocean (Storkey et al., 2018). The CICE configuration includes 5 thickness categories (plus open water), multi-layer thermodynamics and prognostic melt ponds (Ridley et al., 2018). The data assimilation scheme, NEMOVAR (Waters et al., 2015), is used in a 3D-Var FGAT (First Guess at Appropriate Time) configuration to assimilate observations of ocean and sea ice variables.

The observation data types assimilated in the FOAM test system used in this study are the same as for the operational system, namely temperature, salinity, sea level anomaly and SIC (sea ice concentration). No SIT observations are currently assimilated operationally. Sea surface temperature (SST) observations are obtained from ships, moored and drifting buoys, AVHRR sensors onboard NOAA and MetOp satellites, and the VIIRS sensor onboard the Suomi-NPP satellite. Argo floats, moored buoys, gliders, research CTD (Conductivity, Temperature and Depth) instruments and XBTs (Expendable Bathythermographs) provide temperature and salinity profiles. Temperature profiles are also obtained from marine mammals. Sea level anomaly data are provided by altimetry from the Jason-2, Jason-3, Sentinel-3A, CryoSat-2 and AltiKa satellites. SIC measurements from the SSMIS (Special Sensor Microwave Imager/ Sounder) instruments onboard the DMSP series of satellites are assimilated. A control FOAM system using these observations and an experiment system with the additional assimilation of SIT observations derived from CryoSat-2 measurements are employed for this study. Both the SIT assimilation experiment and control systems use a 24-hour assimilation time window and are forced at the surface using hourly wind fields and three-hourly temperature, humidity, precipitation and radiative fluxes from the Met Office NWP system.

The SIT assimilation experiment and control systems have been used to generate daily analysis and 1-day forecasts of ocean and sea ice variables for the three-year period from January 2015 to December 2017. This follows a three-month spin-up period from October 2014, initialised using a previous FOAM reanalysis. The analysis fields were used to initialise 5-day forecasts for each day in selected periods: March - April, June - July, and September - November, for each of the three years in the study. These months were chosen to cover the Arctic ice break-up and freeze-up periods, as well as including March and September to coincide with the annual Arctic maximum and minimum sea ice extents respectively. It should be noted that CryoSat-2 SIT observations are only available between October and April each year, owing to the detrimental impact of summertime melting on the satellite retrievals (Tilling et al., 2016). Therefore June - July was also selected to assess the behaviour of the sea ice forecasts over the summer months when SIT observations are not available for assimilation.

## 2.2 Observations used for SIT assimilation

### 2.2.1 Conversion from freeboard to thickness

The satellite data used in this study are along-track, CryoSat-2 radar altimeter observations of Arctic sea ice freeboard, processed by CPOM (Centre for Polar Observation and Modelling; Tilling et al., 2016). This observation dataset was selected in part as it is available in near-real-time, which is necessary for future implementation into the operational FOAM system. It


would be possible to directly assimilate freeboard observations into the model, rather than converting to SIT first. However, the assimilation of further SIT datasets is planned, including those with direct observations of SIT (e.g. from SMOS). For simplicity, a single additional state vector for the assimilation covering all ice thickness data types was selected. SIT was chosen since it is a model prognostic, whereas freeboard is a diagnostic. Freeboard observations ($f_i$) are converted to SIT ($h_i$) in FOAM
as part of the observation operator (the model run used to compute the model equivalent of the observations), using Eq. 1. Following Tilling et al. (2016) and assuming the ice is floating in hydrostatic equilibrium:

$$h_i = \frac{f_i \rho_w + h_s \rho_s}{(\rho_w - \rho_i)} \tag{1}$$

where $h_s$ is the FOAM modelled snow depth (in m) at the observation time and interpolated to the freeboard observation location. $\rho_w$, $\rho_s$ and $\rho_i$ are densities of water, snow and ice respectively. These are assumed constant, and are set to 1026.0, 330.0 and 917.0 kg m$^{-3}$ respectively, as used in the CICE sea ice model component of FOAM.

Prior to the conversion to SIT, the modelled snow depth is also used to provide a correction to the freeboard observation (the "radar" freeboard; $f_{i\_radar}$) to obtain the "true" or "corrected" freeboard ($f_i$). This accounts for the reduction in speed of
the altimeter radar pulse due to the presence of snow on the sea ice (Tilling et al., 2016). The correction is an addition of the quantity $0.25h_s$ to the radar freeboard observations, based on:

$$f_i = f_{i\_radar} + [(C_o/C_s) - 1.0]h_s = f_{i\_radar} + 0.25h_s \tag{2}$$

where $h_s$ is the snow depth as in Eq. 1, $C_o = 3.0 \times 10^8 \ m \ s^{-1}$ is the speed of light in a vacuum, and $C_s = 2.4 \times 10^8 \ m \ s^{-1}$ is the speed of light in snow.

Currently, CPOM make use of a modified snow depth climatology based on Warren et al. (1999) and halved over first-year ice, for processing CryoSat-2 sea ice freeboard retrievals and conversion to SIT (Tilling et al., 2015). This approach is also used by other centres processing CryoSat-2 freeboard observations: AWI (Alfred Wegener Insitut; Ricker et al., 2014) and
NASA (Kwok and Cunningham, 2015). Instead, here the FOAM modelled snow depth is used, as it has greater spatial and temporal variability than can be obtained from a climatology (figure 1). Using this method also maintains consistency between SIT and snow depth within the model.

Snow depth uncertainty is a large source of error in radar altimetry SIT measurements, both for the retrievals of freeboard and in the subsequent conversion to SIT (e.g. Giles et al., 2007; Ricker et al., 2015). Additional uncertainty is also introduced
through lack of knowledge of the snow and sea ice densities which, although constants in the CICE model used here, are spatially and temporally varying in reality (e.g. Alexandrov et al., 2010; Kern et al., 2015). There are several recent studies on

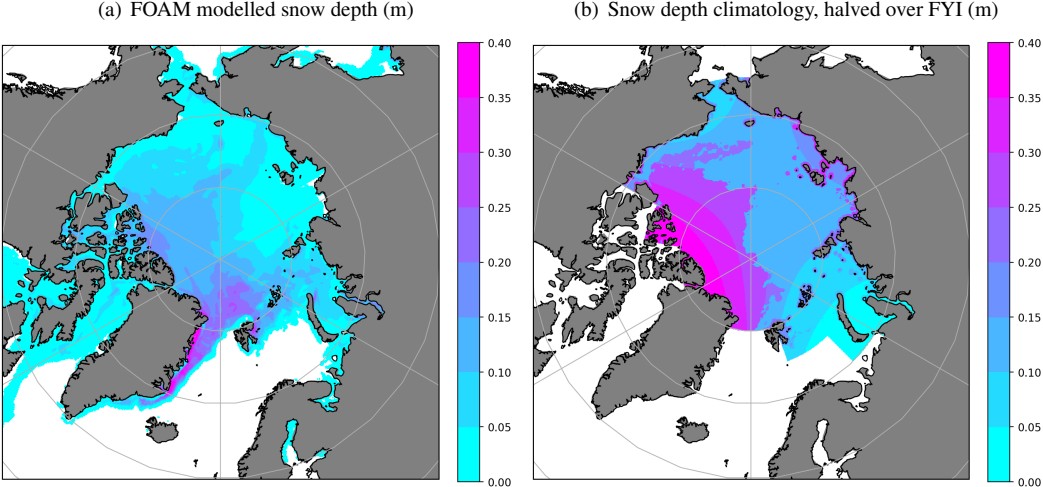

**Figure 1.** Comparison of snow depth from FOAM model and Warren et al. (1999) climatalogy. (a) FOAM daily mean modelled Arctic snow depth (m) for example date 15 January 2015, and (b) Warren et al. (1999) snow depth climatology (m) for January, halved over first year ice (FYI) using EUMETSAT OSI SAF sea ice type observations from 15 January 2015 (Aaboe et al., 2020).

producing daily estimates of snow depth on sea ice, using a variety of methods. These include the use of co-incident CryoSat-2 and ICESat-2 observations (Kwok et al., 2020) and modelling approaches (Petty et al., 2018). Future work will compare the modelled FOAM snow depth to a selection of these datasets. For now, a preliminary validation indicates that the FOAM

snow depth is somewhat thinner than the modified climatology of Warren et al. (1999), as shown on Fig. 1, particularly over multi-year ice.

### 2.2.2    Quality control and pre-processing

Since retrievals of sea ice freeboard from CryoSat-2 are noisy (e.g. Laxon et al., 2013), it is important to apply quality control and pre-processing prior to assimilating the data. As an initial check, following Tilling et al. (2018), freeboard observations

below -0.3 m and above 3 m are rejected, as are any negative values remaining after the conversion to SIT (see Sect. 2.2.1). A Bayesian background check for removal of poor-quality observations (Ingleby and Huddleston, 2005), as currently used for ocean data types in FOAM, was investigated. In this method, any observations that deviate too far from the model background field are rejected, taking into account the model background and observation error variances. However, since the SIT observations are so different from the modelled SIT prior to any SIT assimilation (as will be shown in Sect. 3), it is difficult to

avoid rejecting large numbers of observations when beginning the assimilation. This leads to issues in the analysis quality and excessively long spin-up periods for the assimilation. Use of a background check for subsequent months once the assimilation is established could be investigated further, noting that model drift in the absence of SIT observations over the summer months





may once again lead to the rejection of good quality observations come autumn. Nevertheless, it will be demonstrated in this study that acceptable results can be obtained without including this check.

After the initial gross quality control checks, "super-observations" of the median freeboard observation within a specified radius are created. Super-obbing is often used in data assimilation to subsample satellite observations to the model grid size. It is also an established method of reducing the correlated uncertainty in the observations (e.g. Janjic et al., 2017), which is assumed to be negligible in the assimilation system used here (Sect. 2.4.1). The super-obbing radius was selected to be 10 km, as this is similar to the model grid size in the Arctic on the 1/4 degree tripolar (ORCA025) grid used by NEMO and CICE. The

radius could be increased to include more observations per super-ob, but 10 km allows grid-scale variability to be preserved in the data. For a representative example date of 1 March 2015, the mean number of freeboard observations used to create each super-ob is 18.4 (minimum 2, maximum 101), with the greatest number of observations available at high latitudes where the orbit tracks converge. The mean difference of the observation-minus-background decreased from 0.12 m to 0.08 m pre- and post-superobbing, and the RMSD (root-mean-square difference) from 1.05 m to 0.68 m. This indicates a substantial reduction

of noise in the data.

## 2.3   Observations used for SIT validation

Independent in situ observations suitable for validation of the SIT assimilation experiment and which cover the experiment time period are available from the field campaigns NASA Operation IceBridge (OIB; Kurtz et al., 2013), from airborne electro-magnetic induction (Air-EM) observations as part of PAM-ARCMIP (Pan-Arctic Measurements and Arctic Regional Climate

Model Simulations Project; Haas et al., 2009), and from BGEP (Beaufort Gyre Exploration Project; Krishfield et al., 2014) moorings that measure sea ice draft using upward-looking sonar. The following sections provide details of these datasets.

### 2.3.1   Operation IceBridge (OIB)

The OIB field campaign used an aircraft equipped with a scanning lidar altimeter, a snow radar and cameras to obtain springtime measurements of Arctic sea ice freeboard, thickness and snow depth, for several weeks each year between 2009 and 2018. The

accuracy of the SIT measurements is estimated to be 0.4 m (Farrell et al., 2012). Data for 2015 to 2017 overlap with the dates of the SIT assimilation experiment carried out in this study, and are available for the dates shown in Table 1.

|      | March                  | April   |
|------|------------------------|---------|
| 2015 | 20, 22, 24-31          | 3-4     |
| 2016 | -                      | 20, 28  |
| 2017 | 8-14, 17, 19-20, 24    | 4       |

**Table 1.** Dates of Operation IceBridge data used in this study.

The dataset used here is from the QuickLook V1 product (Kurtz et al., 2019), as the more reliable V2 product was not available for the SIT assimilation experiment period at the time of assessment. The dataset was processed by the OIB project





and comprises 50 km cluster averages of SIT point measurements at a spacing of approximately 25 m. Data from multiple flights
over the same locations were combined into the same cluster for flights fewer than 10 days apart. Only point observations with
low uncertainties were selected for the clusters: less than 1 m uncertainty for very thin ice, up to a maximum uncertainty of 2 m
for ice thicker than 4 m. Clusters were required to have at least 500 point samples, and they have an average of 1670 points with
a maximum of 7000 points. As part of this study, further processing was carried out to remove any cluster observations with
associated standard deviations above 3 m. It is necessary to use spatially averaged rather than point observations for validation,
due to the very high level of noise in the point measurements, and for more appropriate comparison to a model designed to
simulate average sea ice behaviour without sub-gridscale heterogeneity.

Figures 2(a) and (b) show the cluster average SIT observations and standard deviations of the clusters for data from the 2015
- 2017 OIB field campaigns. The figure illustrates the large amount of valuable in situ data available from this project.

### 2.3.2 Airborne electromagnetic combined snow and ice thickness (Air-EM) observations

Field campaigns for PAM-ARCMIP were conducted over the Arctic for several weeks each year between 2001 and 2015. A
helicopter equipped with an instrument known as an "EM-Bird" was used to measure combined SIT and snow depth by airborne
electromagnetic (EM) induction (Haas et al., 2009). Over level ice, the accuracy of Air-EM measurements of combined SIT
and snow depth is 0.1 m (Pfaffling et al., 2007; Haas et al., 2009). Only data for the 2015 field campaign overlap with the dates
of the SIT assimilation experiment in this study, for the following dates: 7-9, 11, 22-23 April.
Similar to the OIB data (Sect. 2.3.1), 50 km cluster averages have been produced by the Air-EM data providers. The clusters
include contributions from multiple flights if these spanned a few days over a small region. Further processing was carried
out as part of this study to remove any cluster observations with standard deviations over 3 m. Figures 2(c) and (d) show the
Air-EM cluster average observations and standard deviations of the clusters for the 2015 field campaign.

### 2.3.3 Beaufort Gyre Exploration Project (BGEP)

The BGEP dataset consists of observations of sea ice draft (thickness of ice below the ocean surface) from upward-looking
sonar (ULS) instruments attached to bottom-anchored moorings in the Beaufort Sea. The locations of the moorings providing
data used in this study are shown on Fig. 2(e). Data are available continuously between September 2003 and August 2017,
with an estimated accuracy of 0.05 - 0.10 m (Krishfield and Proshutinsky, 2006). Data used in this study are monthly means
of sea ice draft, processed by the data providers from the raw 2-second observations, and which do not include any open water
estimates. In the same way as for the other validation datasets, any averaged sea ice draft observations with standard deviations
above 3 m were excluded in this study. Further processing was also undertaken here to convert the measurements of sea ice
draft to SIT by dividing observations by 0.89, following Rothrock et al. (2003).

(a) OIB SIT (m)          (b) OIB standard deviation (m)

(c) Air-EM SIT+snow depth (m)     (d) Air-EM standard deviation (m)

(e) BGEP mooring locations

**Figure 2.** Observations (in m) and standard deviations (in m) of validation data used in this study. (a) Operation IceBridge (OIB) cluster average SIT observations and (b) standard deviations of OIB clusters for 2015 - 2017 field campaigns; (c) Air-EM cluster average observations of combined SIT and snow depth, and (d) standard deviations of Air-EM clusters for 2015 field campaign; (e) locations of BGEP moorings used in this study.





## 2.4 Assimilation of SIT observations

The NEMOVAR assimilation system is used to calculate increments of SIT to be applied to the model, in the same way
as is done for the other assimilated data types in FOAM. The inputs to NEMOVAR are the SIT observations and, for each
observation, the model SIT value aggregated over all thickness categories and interpolated to the location of the observation
at the closest model time-step to the observation time, during a one-day model forecast. The uncertainties associated with the
observations and model forecast are also provided to NEMOVAR, in the form of error covariances, as described in the following
subsections. NEMOVAR outputs the changes required to bring the model SIT into line with that day's observations, taking into
account their respective uncertainties, as a field of increments on the model grid. These SIT increments are applied to the CICE
model using an Incremental Analysis Update (IAU) method (Bloom et al., 1996), as is used for the other variables in FOAM.
In this method, a fraction of the increments is added at each time-step during a 24-hour period, such that the total increment
is applied by the end of the model day. Following Blockley and Peterson (2018), SIT increments are added to each of the five
sub-grid SIT categories in proportion to the initial volume distribution of ice across each category, if the ice concentration
within that category is above 1%. Changes resulting from SIT increments are only made where the grid cell aggregate SIC is
greater than a conservatively-chosen value of 40%. This means that only SIC increments are able to add new ice, since this data
is deemed the more reliable, particularly for the generally thinner ice of the marginal ice zone. The SIT increments are applied
after any SIC assimilation changes at each time-step and, similar to SIC, no balancing is performed with the other variables.

### 2.4.1 Observation error covariance

The assimilation system requires estimates of observation uncertainties for SIT. The higher the magnitude of the observation
uncertainty, the smaller the impact the observation will have on the analysis. Observation uncertainties are provided as obser-
vation error variances (OBE), with the assumption that the observation error is uncorrelated. This is not necessarily true, but is
a standard simplification (e.g. Stonebridge et al., 2018). The OBE is a combination of measurement uncertainty and representa-
tion uncertainty. Measurement uncertainty includes the raw measurement error and uncertainties due to the retrieval algorithm,
and representation uncertainty results from unresolved scales and processes in the model, observation operator uncertainty
and quality control uncertainty (e.g. Janjic et al., 2017). In this study, the representation uncertainty is set to 0.05 m standard
deviation as an initial estimate, being of a similar magnitude to the minimum measurement uncertainty (see below). Further
work is required to refine this specification. Since estimates of measurement uncertainty are not provided with the CPOM data
used in this study, this has instead been determined using a simple function based on Fig. 2 of Ricker et al. (2017), who derived
relationships between the magnitude of CryoSat-2 ice thicknesses and their associated measurement uncertainties, in terms of
% error. Further details of these methods are given in Ricker et al. (2014). The SIT and uncertainty relationships determined
by Ricker et al. (2017) have slightly different characteristics at different times of the year, but using a single, general function





in this study as a first parameterisation of the measurement uncertainty is a reasonable choice. The function used here is shown on Fig. 3(a), and is specified as follows, in m for thickness $h_i$ and standard deviation $\sigma$, with a cap of 8 m:

$$
\quad \sigma = \begin{cases} 8 & \text{for } h_i < 0.7\text{m} \\ [7e(\frac{-1}{0.3-h_i^e}) + 1] * \frac{h_i}{100} & \text{for } 0.7 \leq h_i < 3.0\text{m} \\ \{[(h_i - 3.0) * 5] + [7e(\frac{-1}{0.3-3.0^e}) + 1]\} * \frac{h_i}{100} & \text{for } h_i \geq 3.0\text{m} \end{cases} \quad (3)
$$

The uncertainty standard deviation associated with each SIT observation is calculated using the function given in Eq. 3. CryoSat-2 freeboard retrievals over sea ice less than around 1 m thick have particularly large uncertainties, owing to the small difference in elevation between the ocean surface and the ice surface above (Ricker et al., 2017). The uncertainty estimate for these observations has therefore been set to a very high percentage of the observation magnitude, ensuring these data do not

influence the analysis. For SIT between ∼1.5-3 m the uncertainty is at its minimum, meaning these observations are given stronger weighting in the analysis, and uncertainty begins to increase again for observations thicker than 3 m. Sensitivity tests were conducted to tune the final form of the function and to set the minimum uncertainty assigned to the most reliable observations in order to produce the optimum SIT analysis results. Figures 3(b) and (c) show the monthly mean observed CryoSat-2 SIT for March 2015, and the measurement uncertainty standard deviation for the data, derived from the parameterisation shown

in Fig. 3(a), and both binned onto a 1/4 degree regular grid. The figures show the largest uncertainties occuring in regions of very thick ice, as there are limited observations of thin ice in the dataset following the gross error check (Sect. 2.2.2).

There is a high level of random uncertainty in the CryoSat-2 along-track observations owing to speckle, sea ice roughness, sea surface height measurement errors and the variation in densities of snow and ice (Ricker et al., 2014; Landy et al., 2020), and the OBE estimate does not explicitly take this into account. These random uncertainties can be greater than 1 m, and

are usually removed by extensive averaging (e.g. Ricker et al., 2017). The super-obbing process described in Sect. 2.2.2 will mitigate this issue somewhat, but the uncertainty in the observations will be higher around the ice edge since there are fewer overlapping orbits at these latitudes. This results in as few as 2 observations in a super-observation, compared with up to 100 or so at higher latitudes. Work is currently underway to also assimilate SMOS SIT data to improve the representation of thin ice in FOAM, and this will improve the analysis in the regions most affected by this issue. Other possible improvements include

filtering out the affected observations from the analysis by increasing the OBE in the CryoSat-2 data below a specified latitude or below a minimum number of observations used in the super-observation. Additionally, a check of the observation quality against the model SIT background could be used, although, as discussed in Sect. 2.2.2, this method relies on the model being unbiased.

### 2.4.2 Background error covariance

Uncertainties in the 1-day model forecast prior to the assimilation of observations (the "background") are parameterised for the SIT assimilation as spatially and seasonally-varying fields of background error variances (BGE) with associated spatial error correlation length scales. The seasonal BGE variances are interpolated to the model date within the FOAM system. This

(a) SIT measurement uncertainty parameterisation

(b) Observed SIT (m)   (c) SIT measurement uncertainty standard deviation (m)

**Figure 3.** Observation uncertainty estimates for SIT. (a) Parameterisation for SIT measurement uncertainty in terms of standard deviation (m), as a function of SIT (m). March 2015 mean binned (b) observed CryoSat-2 SIT (m) and (c) observation measurement uncertainty standard deviation (m), calculated using the parameterisation shown in (a), for the observations shown in (b). Observations have undergone initial quality control and pre-processing.



method is also used for SIC assimilation in FOAM (Blockley et al., 2014). For SIT, the BGE for each season and the correlation length scale were estimated using the "Canadian Quick" covariance method (Polavarapu et al., 2005) with 3 years of FOAM

hindcast SIT data (1 June 2015 to 31 May 2018). In this method, differences between daily model fields are used as a proxy for the model forecast error. The spatial correlations in the results were assessed and yielded an estimate of 50 km for the minimum SIT correlation length scale. This value was used as a constant length scale everywhere, except at very high northern latitudes where it was extended to compensate for the data gap north of 88.0 N owing to the orbital inclination of the CryoSat-2 satellite (the "pole hole"). The length scale was increased to 100 km for observations at latitudes north of 87.5 N, making the

assumption that the model uncertainties near the pole are highly correlated with the uncertainties at 87.5 N. This has the effect of filling the pole hole using increments spread from the surrounding area, to allow the SIT analysis to vary in this region. A sensitivity test was used to select the magnitude of the extended length scale and the latitude threshold. The values chosen appear to give a good result (Fig. 4): the pole hole is filled, but with minimal impact on the increments nearby. A quantitative validation is not possible, owing to the absence of satellite or in situ SIT observations at this location.

Figure 4 also demonstrates that the 50 km length scale allows information from the observations to propagate spatially over the domain, which is helpful owing to the sparseness of the daily CryoSat-2 observations in regions where the orbit tracks do not overlap (Fig. 4(d)). A dual length scale correlation formulation is available in NEMOVAR (Mirouze et al., 2016; Fiedler et al., 2019) and use of this for SIT remains an avenue for future investigation.

    SIT modelled by the FOAM system is too thin without SIT assimilation (as will be shown in Sect. 3), so the model BGE

calculated using the Canadian Quick method is likely to be an underestimate. However, it provides a starting point for iterations of BGE calculations using different methods, once the SIT assimilation is in place. Examples of such methods are given in Bannister (2008) and include those based on innovation (observation-minus-background) correlations (Hollingsworth and Lönnberg, 1986), differences between forecasts of varying lengths (the "NMC" method; Parrish and Derber, 1992), or ensemble data assimilation methods (Houtekamer et al., 1996).

## 3   SIT assimilation results

SIT derived from along-track CryoSat-2 freeboard observations can be successfully assimilated into the FOAM forecasting system. Figures 5(a) and (b) show daily mean SIT analysis fields for the SIT assimilation experiment and the control respectively, for 15 January 2015 as an example date. Gridded CPOM CryoSat-2 SIT observations for January 2015 are also shown for comparison (Fig. 5(c)). The results demonstrate that the FOAM system is able to reproduce the monthly gridded obser-

vation field by assimilating daily along-track data. The control SIT model field is smoother than the observations, and much thinner in the Canadian Arctic region. The SIT assimilation experiment has produced a thicker sea ice field, particularly in the Canadian Arctic and around Greenland, and introduced more spatial variability throughout.

    Figure 5(a) demonstrates that the background error correlation length scale for the SIT assimilation is suitable, since the satellite tracks of the day's observations are not obvious in the analysis field (length scale too short), and there is no excessive

smoothing (length scale too long). Despite the absence of observations, the pole hole has a realistic-looking SIT, which is

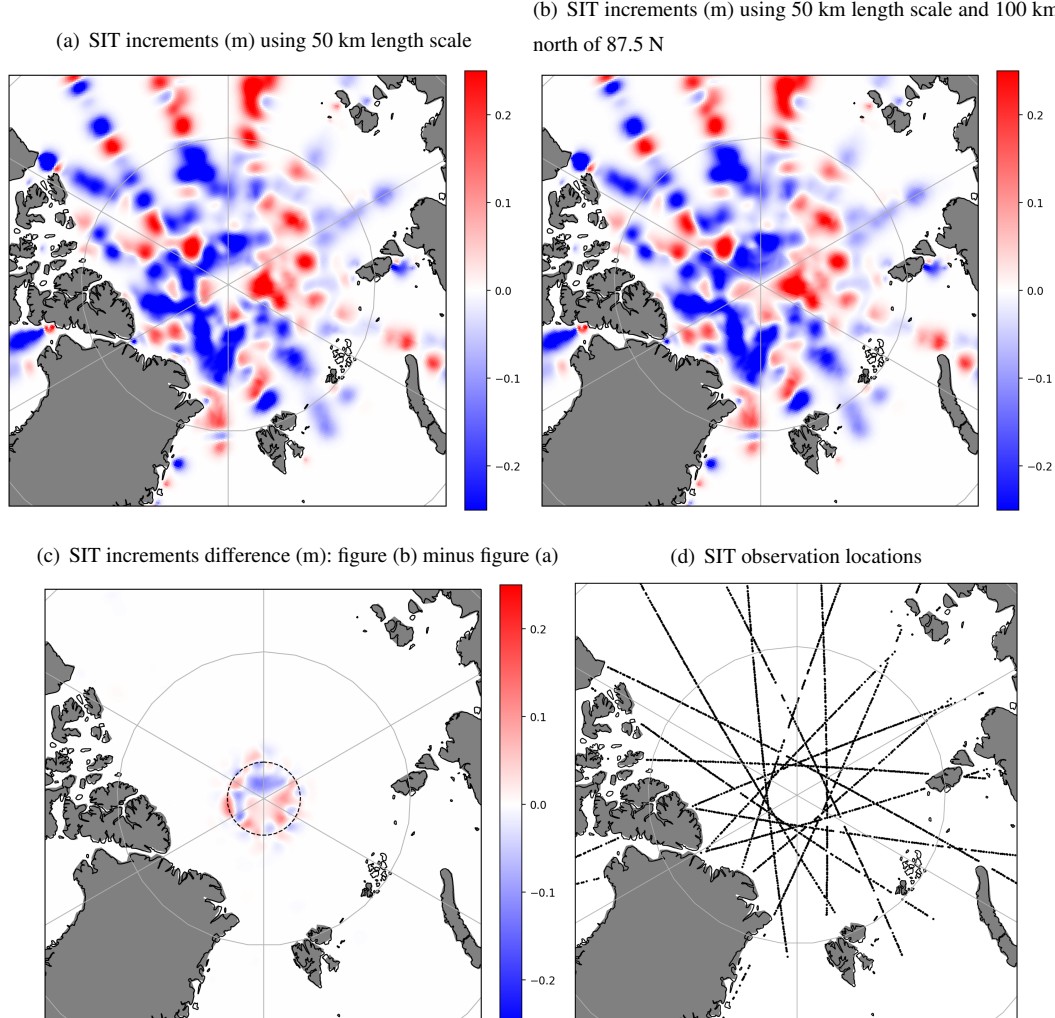

**Figure 4.** Demonstrating filling of the satellite observation data gap at the north pole using SIT information from the surrounding area. SIT increments (m) produced using different background error correlation length scales, example date 30 January 2015 for (a) 50 km length scale throughout, (b) 50 km length scale, with 100 km north of 87.5 N, and (c) difference between (a) and (b), with 87.5 N gridline shown as a dashed black line. (d) Locations of assimilated SIT observations for the domain shown, 30 January 2015.

thicker than in the control (Fig. 5(b)). This demonstrates that the method of using SIT information spread from the surrounding area is successful.

Figure 6 shows the difference between monthly averaged SIT fields for the SIT assimilation experiment minus the control, for the months of the Arctic sea ice maximum and minimum extents, March and September respectively. 2016 is shown as an example year. The main impact of the SIT assimilation in March is to increase the SIT in the Canadian Arctic and European





sectors (Fig. 6(b)). Some thinning of the sea ice is also seen outside of these regions, which for March 2016 occurs around the East Siberian, Chukchi and Laptev Seas. Although there are no SIT observations available for assimilation between May and September inclusive each year, differences between the SIT assimilation experiment and control are still apparent in the September SIT field (Fig. 6(b)). This demonstrates that the effect of the SIT assimilation on the model in the months prior to

May persists throughout the summertime to September. This effect was also shown in the experiments of Blockley and Peterson (2018) and Allard et al. (2018), and illustrates that the SIT assimilation is having the desired impact on the FOAM model. The SIT assimilation has a minimal impact on the SIC model field in March and September (not shown), with small changes in concentration seen mostly around the ice edge at times of maximum ice extent and within the pack when the minimum extent occurs.

Differences between SIT analyses and forecasts generated by the SIT assimilation experiment and the control are assessed in detail in the following sections, using validation statistics from the assimilation system and comparisons with independent in situ observations.

## 4   Validation using assimilation statistics

The performance of the FOAM SIT and SIC 1-day model forecasts (also referred to as the backgrounds onto which the

assimilation increments are added) has been assessed using CryoSat-2 SIT and SSMIS SIC (Tonboe et al., 2017) observations, prior to them being assimilated. The performance of the FOAM 5-day forecasts of SIT and SIC has also been assessed using the same observations, during the months for which these longer forecasts were produced. Matchups between the forecasts and observations were obtained by interpolating the daily-mean model forecast fields to the observation locations.

Figure 7 shows timeseries of SIT observation-minus-background daily mean difference and RMSD (root-mean-square dif-

ference) for the SIT assimilation experiment and the control, for January 2015 to December 2017. Note that there are no CryoSat-2 SIT observations, and hence statistics, available between May and September of each year. The improvement in both the mean difference and RMSD is clear, with substantial reductions for the SIT assimilation experiment compared to the control of 0.46 m mean difference (0.33 m RMSD) for March-April (ice break-up), and 0.75 m mean difference (0.41 m RMSD) for October-November (ice freeze-up), averaged over the Arctic for 2015 - 2017. Figure 7 also illustrates that the

benefit to the model from the SIT assimilation continues throughout the summer months since, despite quite an increase, the differences to the observations still remain lower than in the control when the statistics become available again in October. Additionally, there is a very quick reduction in the model differences to the SIT observations once the assimilation restarts.

Figures 8 and 9 show the same statistics as Fig. 7, but as spatially-binned plots of mean difference and RMSD for March-April and October-November 2015, to examine the regional variation in results for the ice break-up and freeze-up periods

respectively. The improvement (reduction) in the mean difference and RMSD due to SIT assimilation compared to the control is seen over large areas of the Arctic, both at the start of the melt period (Fig. 8, though some differences still remain in the regions of thickest ice, in the Canadian Arctic and along the eastern coast of Greenland), and more substantially during the

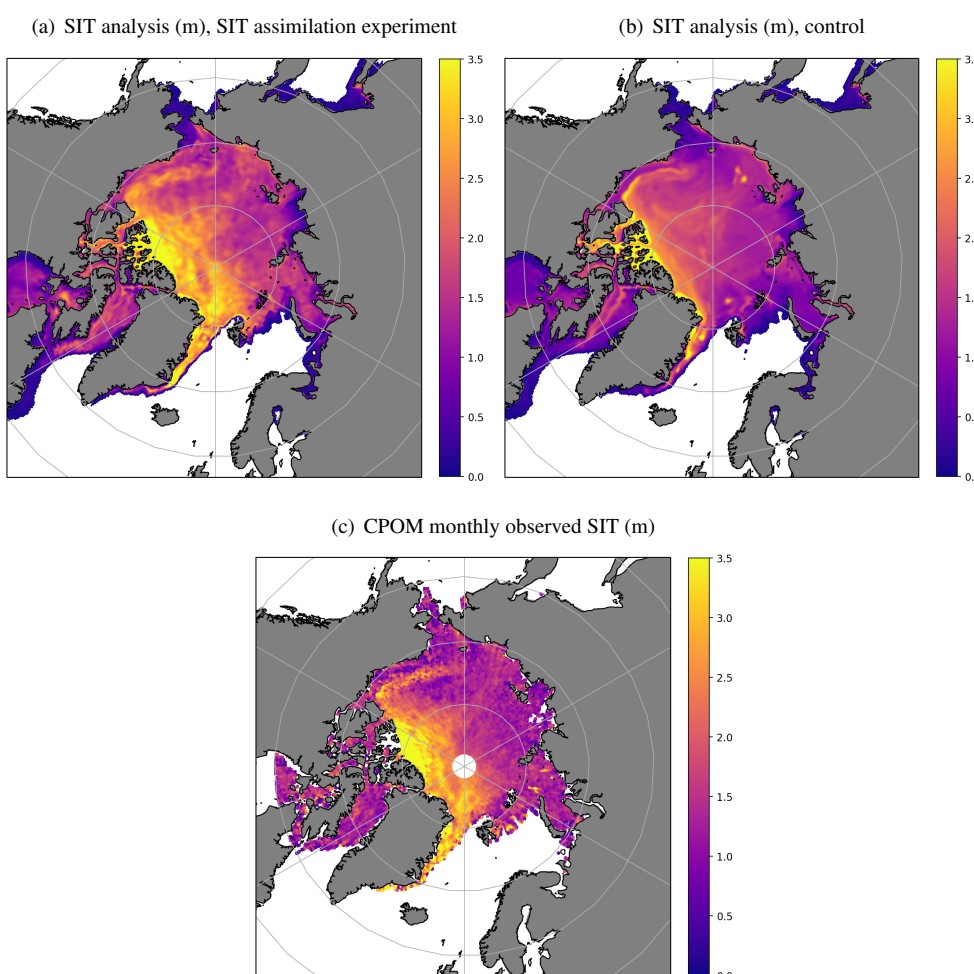

**Figure 5.** Modelled and observed SIT (in m). Daily mean FOAM SIT analysis for (a) SIT assimilation experiment and (b) control, 15 January 2015, and (c) 5 km gridded CPOM CryoSat-2 SIT observations, January 2015.



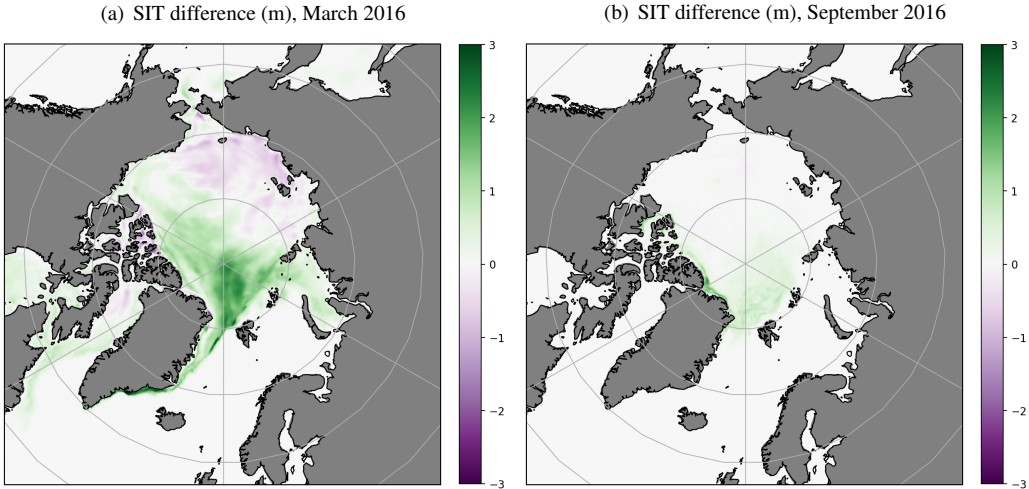

**Figure 6.** Monthly mean SIT differences (in m) for SIT assimilation experiment minus control, March and September 2016. Green (purple) indicates thicker (thinner) ice in the SIT assimilation experiment compared to the control.

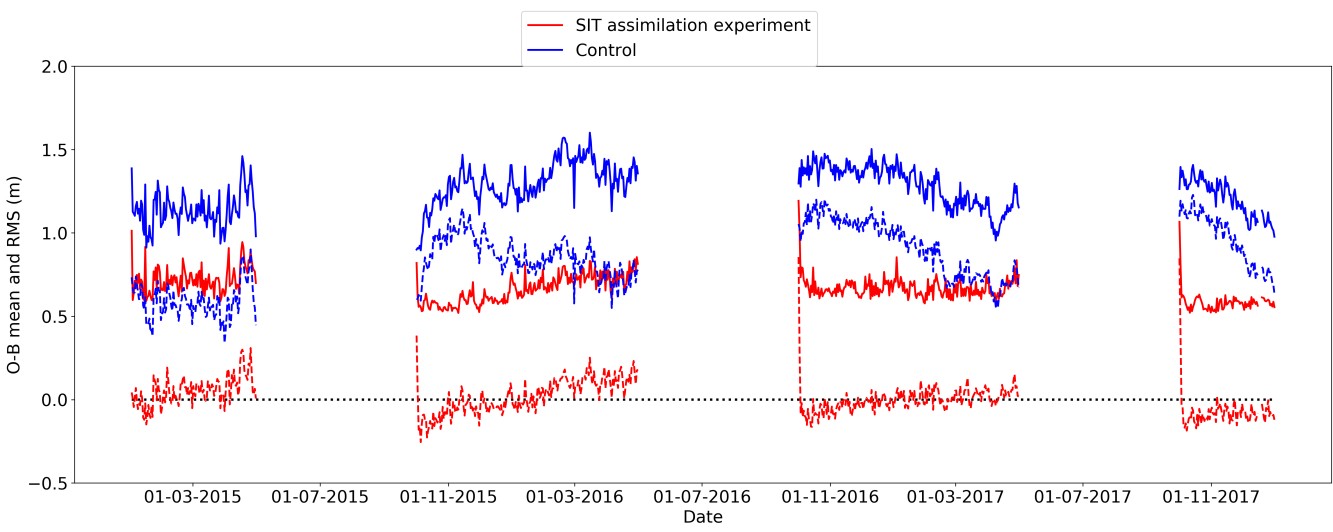

**Figure 7.** Timeseries of daily SIT observation-minus-background (O-B) statistics for January 2015 to December 2017. Mean difference is broken line, RMSD is solid line. 0.0 m reference line shown in dotted black.





freeze-up (Fig. 9). This effect is also seen in 2016 and 2017 (not shown). The largest regional improvements are in March-April, where the mean difference is above 1.30 m and 1.22 m for the RMSD.

355 Figure 10 shows spatially-binned plots of SIC observation-minus-background mean difference and RMSD, for July 2015. The difference between the SIC results for the SIT assimilation experiment and control are negligible for the other months. Figure 10 illustrates that both the mean difference and RMSD for SIC are improved (reduced) in the Atlantic sector by around 0.05 sea ice fraction for the SIT assimilation experiment compared to the control. This occurs despite the assimilation of SIT ceasing at the end of April when CryoSat-2 data production is suspended over the summer months, and is likely to be due to

360 the assimilation introducing thicker ice that melts more slowly than in the control, leading to improvements in the SIC field. This effect ceases to become important by early August and does not appear to affect SIC in the freeze-up period (not shown). The effect is also seen, though to a lesser extent, in 2016 and 2017 (not shown).

 Figure 11 shows validation statistics for FOAM 1- to 5-day forecasts of SIT and SIC compared to observations, for the SIT assimilation experiment and the control. The break-up (March-April) and freeze-up (October-November) periods in the Arctic

365 are shown for SIT and summertime (July) for SIC, for 2015 as an example year. The benefit of the CryoSat-2 assimilation on the forecasts of SIT is clear, with substantial improvements seen in both the mean difference and RMSD (Figs. 11(a) and (b)). The improvement compared to the control is particularly marked for the freeze-up period (Fig. 11(b)), as was illustrated on Fig. 9 for the 1-day forecast (assimilation background). The uncertainty in the forecasts themselves grows by only a few centimetres over the 5-day period, as the thick sea ice observed by CryoSat-2 does not change rapidly on this timescale.

370 Figure 11(b) also unexpectedly shows that there is a slight improvement in the mean difference (becomes closer to zero) of the observations minus the model over the 5-day forecast in the freeze-up period, for the SIT assimilation experiment. The improvement may be due to smoothing out over the 5-day period of spatial noise introduced by the assimilation. Noise in the assimilation could be improved by further tuning of the error covariances.

 Figure 11(c) shows SIC forecast statistics for July 2015 (no SIT observations are available for this time period), and demon-

375 strates improvement in the summertime SIC forecasts for the SIT assimilation experiment compared to the control. This was highlighted above for assimilation background (1-day forecast) assessment results in July (Fig. 10), which demonstrated the improvement is located in the European sector. As discussed, this is likely due to the retention of thicker ice from the SIT assimilation earlier in the year, which reduces the forecast error growth in SIC during the summer. Changes in mean difference and RMSD statistics for SIC forecasts generated by the SIT assimilation experiment and control at other times of the year are

380 negligible.

## 5 Validation using independent in situ observations

FOAM model output from the SIT assimilation experiment and the control was validated using airborne radar and laser altimeter observations of SIT from NASA OIB (Operation IceBridge), moored upward-looking sonar observations of sea ice draft from BGEP (Beaufort Gyre Exploration Project), and combined SIT and snow depth observations from PAM-ARCMIP

385 (Pan-Arctic Measurements and Arctic Regional Climate Model Simulations Project) using airborne electromagnetic induction

(a) Mean difference (m), SIT assimilation experiment

(b) Mean difference (m), control

(c) RMSD (m), SIT assimilation experiment

(d) RMSD (m), control

**Figure 8.** Spatially-binned SIT observation-minus-background statistics (in m) for March-April 2015. For (a), (b): Green (purple) indicates that model ice is thinner (thicker) than the observations. For (c), (d): A reduced RMSD indicates an improvement in the model compared to the observations.



(a) Mean difference (m), SIT assimilation experiment

(b) Mean difference (m), control

(c) RMSD (m), SIT assimilation experiment

(d) RMSD (m), control

**Figure 9.** Spatially-binned SIT observation-minus-background statistics (in m) for October-November 2015. For (a), (b): Green (purple) indicates that model ice is thinner (thicker) than the observations. For (c), (d): A reduced RMSD indicates an improvement in the model compared to the observations.



**Figure 10.** Spatially-binned SIC observation-minus-background statistics (in sea ice fraction) for July 2015. For (a), (b): Green (purple) indicates that model ice concentration is lower (higher) than the observations. For (c), (d): A reduced RMSD indicates an improvement in the model compared to the observations.

(a) SIT, March-April 2015

(b) SIT, October-November 2015

(c) SIC, July 2015

**Figure 11.** Mean difference (broken lines) and RMSD (solid lines) of SIT (in m) and SIC (in fraction) observations minus 1- to 5-day forecasts for (a) SIT in March-April 2015 (break-up period), (b) SIT in October-November 2015 (freeze-up period), and (c) SIC in July 2015. 0.0 m reference lines shown in dotted black.





(Air-EM). Details of these datasets and pre-processing applied are given in Sect. 2.3. The CryoSat-2 SIT observations used in the SIT assimilation experiment were also compared with these in situ datasets, to provide context for the changes in the model due to the assimilation.

In order to produce matchups between FOAM model fields and independent in situ validation observations, the model fields were interpolated to the observation locations for each date. An offline observation operator, part of the NEMOVAR assimilation code (Waters et al., 2015), was used for this purpose. The OIB and Air-EM data are provided as daily means, so the daily mean model field for the date of the observations was used when producing the matchups. For the monthly BGEP data, the monthly mean of the model fields was used.

For matchups of CryoSat-2 SIT observations with the in situ data, 30 days of the quality-controlled, super-obbed, assimilated subset of the full dataset was gridded onto the same 1/4 degree tripolar grid (ORCA025) used by the model. The gridded observation field was then interpolated to the observation locations. The 30-day periods were chosen to cover the time period of each yearly field campaign for the OIB and Air-EM validation datasets, and calendar months were used when producing matchups with the BGEP dataset. For comparison to the combined SIT and snow depth of the Air-EM observations, the 30-day mean of the modelled snow depth was added to the gridded CryoSat-2 SIT observations before producing the matchups. The snow depth from the SIT assimilation experiment rather than the control was used, to maintain consistency with the assimilated SIT observations.

## 5.1 Operation IceBridge (OIB)

Figures 12(a) and (b) show OIB cluster average SIT observations from 2015, 2016 and 2017 minus the 5-day SIT forecast produced by the FOAM system for the SIT assimilation experiment and the control, respectively. The figures demonstrate that overall, the SIT assimilation experiment produces a substantial improvement in the SIT forecast field compared to the control. Results for validation of the SIT analysis (a 1-day forecast that has been corrected by the assimilated observations, and is used to initialise subsequent forecasts; not shown) are similar to those for the 5-day forecast.

| | SIT assimilation experiment | | Control | | CryoSat-2 obs |
|---|---|---|---|---|---|
| | Analysis | 5-day forecast | Analysis | 5-day forecast | |
| Correlation coefficient | 0.76 | 0.80 | 0.57 | 0.58 | 0.78 |
| Mean difference | 0.14 | 0.15 | 0.75 | 0.74 | 0.07 |
| Absolute mean difference | 0.51 | 0.48 | 0.87 | 0.86 | 0.49 |
| RMS of differences | 0.69 | 0.65 | 1.11 | 1.11 | 0.65 |
| Standard deviation of differences | 0.68 | 0.64 | 0.82 | 0.82 | 0.65 |

**Table 2.** Statistics for all matchups (total 547) of OIB SIT observations from 2015-2017 with FOAM SIT analysis, FOAM 5-day SIT forecast, and assimilated CryoSat-2 SIT observations. Differences are in situ observation minus model or satellite observation, in m.

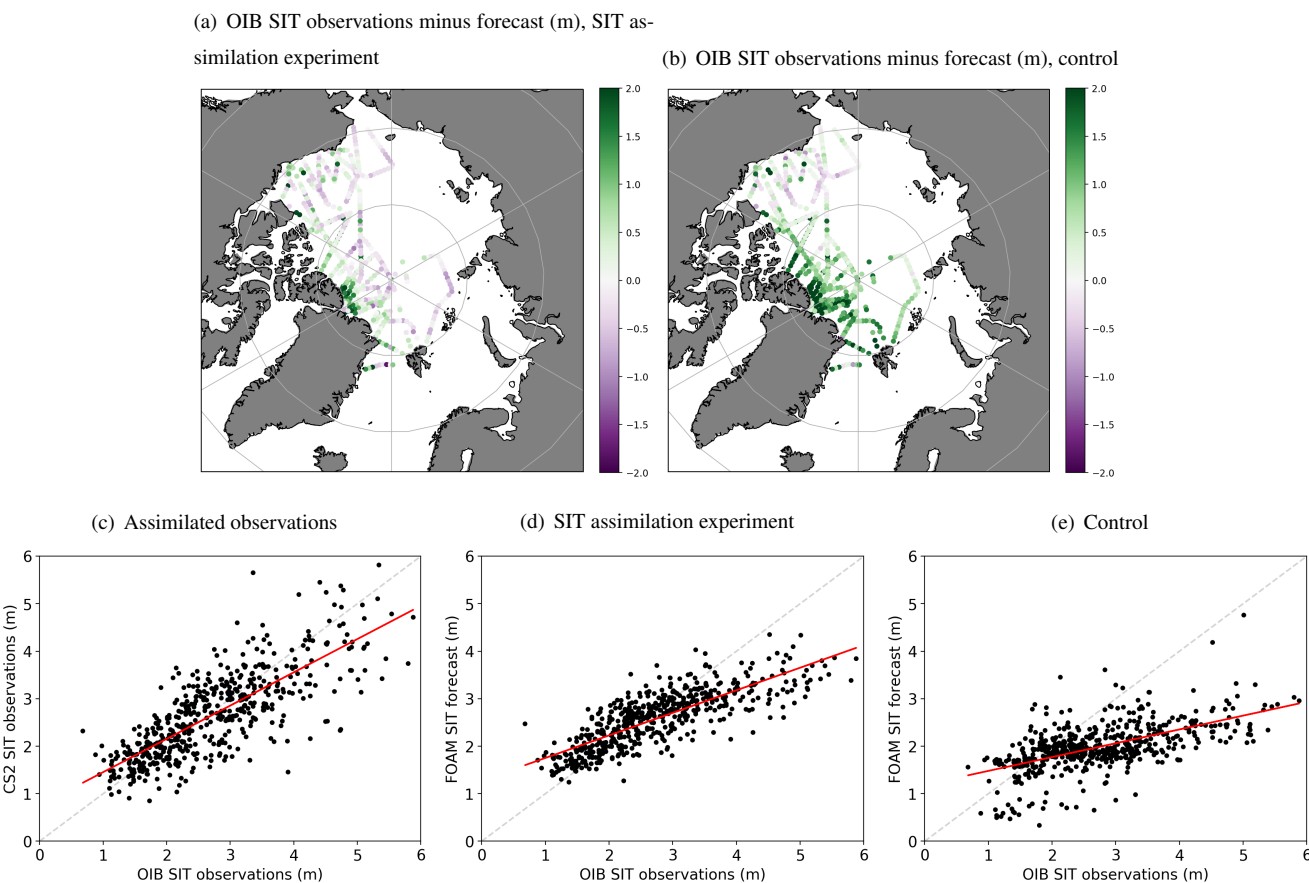

**Figure 12.** Validation of CryoSat-2 (CS2) SIT observations and FOAM 5-day SIT forecasts against Operation IceBridge (OIB) cluster average SIT observations from 2015-2017. OIB observations minus FOAM forecasts (in m) for (a) SIT assimilation experiment and (b) control. Green (purple) in (a), (b) indicates the model ice is thinner (thicker) than the observations. Scatter plots of OIB observations (in m) with (c) monthly, gridded, quality-controlled CryoSat-2 SIT observations (in m); and FOAM forecasts (in m) for (d) SIT assimilation experiment and (e) control. Best fit lines for (c), (d) and (e) are shown in red, with a 1:1 reference line in broken grey.





Figure 12(c) shows a scatter plot of matchups between CryoSat-2 SIT observations and OIB cluster average SIT observations for 2015, 2016 and 2017. Summary statistics of the relationships are given in Table 2. These results demonstrate that, using OIB

SIT as a reference for validation, the CryoSat-2 SIT observations are more reliable than the model without SIT assimilation. Furthermore, the results show that assimilating these observations leads to improvements in the model performance.

Figures 12(d) and (e) show scatter plots of OIB cluster average SIT observations for 2015, 2016 and 2017, along with the corresponding FOAM 5-day SIT forecast matchup points, for the SIT assimilation experiment and control respectively. These figures illustrate the improvement in the relationship between the validation observations and the FOAM fields for the SIT

assimilation experiment compared to the control, with fewer outlying points and the best fit line being closer to the ideal 1:1 line. The statistical relationships between the observations and FOAM 5-day SIT forecasts, as well as the FOAM SIT analysis, are summarised in Table 2. It is demonstrated that there are improvements across all statistics for the SIT assimilation experiment compared to the control. As indicated in Sect. 4 using assimilation statistics, validation against the independent OIB observations also demonstrates an unexpected slight improvement in the 5-day forecast compared with the analysis (Table 2).

As discussed, this may indicate spatial noise is introduced by the assimilation, which is smoothed out over the 5-day forecast period. The good results for the forecast also demonstrate that, throughout the 5-day forecast period, the model is able to successfully retain improvements to the SIT field introduced by the initial assimilation. This was also demonstrated using assimilation statistics for validation in Sect. 4. There is very little difference between the statistics for the SIT analysis and 5-day forecast for the control run, since the control analysis has not assimilated SIT data and the generally thick ice being

assessed changes slowly over this timescale.

## 5.2 Airborne electromagnetic combined snow and ice thickness (Air-EM) observations

Figures 13(a) and (b) show Air-EM cluster average observations of SIT plus snow depth from 2015 minus FOAM 5-day forecasts of the same quantity, for the SIT assimilation experiment and control respectively. For observations in the Canadian Arctic, it can be seen that differences between the Air-EM observations and the FOAM output are generally reduced for

the forecast runs assimilating SIT observations compared to the control. However, there are some larger negative differences, which indicate that the modelled combined SIT and snow depth in this region has increased too much for the SIT assimilation experiment. Validation of the FOAM SIT analysis against Air-EM observations (not shown) yields similar results to the forecasts.

Figures 13(a) and (b) also show that differences between the observations and the model in the Beaufort Sea are smaller for

the control than for the SIT assimilation experiment, indicating the assimilation leads to a degradation in model performance for this region. However, it should be noted that there are several OIB observations in this sector (Sect. 2.3.1) and these agree much better with the model output than do the Air-EM observations. This potentially indicates an uncertainty in the quality of the Air-EM observations, which are also fewer in number than the OIB data.

Figure 13 also shows scatter plots of the Air-EM cluster average observations of combined SIT and snow depth from 2015,

with CryoSat-2 SIT observations plus modelled snow depth (Fig. 13(c)), and FOAM 5-day forecast matchups of SIT plus snow



depth for the SIT assimilation experiment and control (Figs. 13(d),(e) respectively). Summary statistics of the relationships between these datasets and also with the FOAM SIT analysis plus snow depth are given in Table 3.

Figure 13(c) and Table 3 indicate that the CryoSat-2 SIT plus snow depth observations are substantially thicker than the Air-EM observations. Since the model SIT is too thin, assimilating the SIT observations has the compensating effect of reducing
the mean difference of the model compared to the Air-EM observations, and this also reduces the RMSD (Table 3). However, the model standard deviation and correlation coefficient are poorer on assimilation of this data. Uncertainty in the snow depth will be contributing to this issue, although the difference between the CryoSat-2 and Air-EM observations is greater than the snow depth itself.

Figure 14 shows the difference in snow depth between the SIT assimilation experiment and the control, for 11 April 2015
as an example date when Air-EM and FOAM matchups are available. This illustrates that the snow is generally ∼5 cm deeper at the Air-EM matchup locations (shown on Figs. 13(a) and (b)) for the SIT assimilation experiment than for the control. This indicates that uncertainties in the modelled snow depth will be affecting results, not only for the Air-EM and CryoSat-2 comparisons, but for assessment of the SIT assimilation experiment and the control using Air-EM validation data. An assessment of the snow mass budget in the model shows that the deeper modelled snow in the SIT assimilation experiment is due mostly
to the thicker ice from the assimilation reducing the ice surface temperature, which results in reduced evaporation/sublimation compared to the control. Note that this change, despite increasing the mean difference between the CryoSat-2 and Air-EM matchups, actually brings the modelled snow depth closer to climatology (see Fig. 1).

|  | SIT assimilation experiment | | Control | | CryoSat-2 obs |
|---|---|---|---|---|---|
|  | Analysis | 5-day forecast | Analysis | 5-day forecast |  |
| Correlation coefficient | 0.68 | 0.69 | 0.82 | 0.82 | 0.68 |
| Mean difference | -0.03 | 0.04 | 0.48 | 0.48 | -0.67 |
| Absolute mean difference | 0.46 | 0.45 | 0.58 | 0.57 | 1.03 |
| RMS of differences | 0.55 | 0.54 | 0.67 | 0.67 | 1.25 |
| Standard deviation of differences | 0.55 | 0.54 | 0.48 | 0.47 | 1.06 |

**Table 3.** Statistics for all matchups (total 45) of Air-EM SIT plus snow depth observations from 2015 with FOAM SIT analysis plus snow depth, FOAM SIT plus snow depth 5-day forecast, and assimilated CryoSat-2 SIT observations plus snow depth. Differences are in situ observation minus model or satellite observation, in m.

Unlike for the validation against OIB observations (Sect. 5.1, Table 2), the 5-day forecast statistics do not show an improvement over the analysis for the SIT assimilation experiment (Table 3). This may be a result of uncertainty due to snow depth
in the Air-EM validation, a potential sampling error owing to the limited number of Air-EM matchups, or uncertainties in the Air-EM observations themselves.



(a) Air-EM SIT+snow observations minus forecast (m), SIT assimilation experiment

(b) Air-EM SIT+snow observations minus forecast (m), control

(c) Assimilated observations plus snow

(d) SIT assimilation experiment

(e) Control

**Figure 13.** Validation of CryoSat-2 (CS2) SIT plus snow depth observations and FOAM 5-day SIT plus snow depth forecasts against Air-EM cluster average combined SIT and snow depth observations from 2015. Air-EM observations minus FOAM forecasts (in m) for (a) SIT assimilation experiment and (b) control. Green (purple) in (a), (b) indicates the model ice is thinner (thicker) than the observations. Scatter plots of Air-EM observations (in m) with (c) monthly, gridded, quality-controlled CryoSat-2 SIT observations plus modelled snow depth (in m); and FOAM forecasts (in m) for (d) SIT assimilation experiment and (e) control. Best fit lines for (c), (d) and (e) are shown in red, with a 1:1 reference line in broken grey.

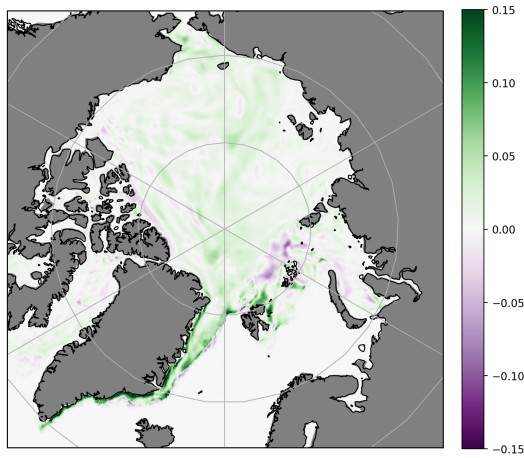

**Figure 14.** Difference in FOAM daily mean modelled snow depth (m) for SIT assimilation experiment minus control, 11 April 2015.

## 5.3 Beaufort Gyre Exploration Project (BGEP)

Figure 15 shows scatter plots of monthly mean SIT observations derived from BGEP moorings for January 2015 to August 2017 (no BGEP data were available after this date), with CryoSat-2 SIT observations and FOAM SIT analysis matchups.
Results for FOAM 5-day forecasts are not shown since these were only produced for selected months, and the limited number of data points would lead to unreliable statistics. Table 4 summarises the statistical relationships between the observation and model or satellite datasets. Results in Fig. 15 and Table 4 are separated into bins of thin ice (below 1 m) and thicker ice (equal to or above 1 m), based on the BGEP SIT. Points on Fig. 15 are colour-coded by season, illustrating the close relationship between SIT and time of year.

Figure 15(a) and Table 4 demonstrate that CryoSat-2 SIT observations over 1 m thick are marginally less reliable than the model without assimilation, using the BGEP data as a validation reference. Consequently, statistics for the SIT assimilation experiment are slightly poorer than for the control (Figs. 15(b),(c) and Table 4). However, overall, for SIT above 1 m, the control, SIT assimilation experiment and CryoSat-2 statistics are good, and the difference is considerably smaller than for observations below 1 m. It is therefore less likely that a dramatic improvement could be achieved for this region by the
assimilation of thick SIT observations from CryoSat-2, unlike for other areas such as the Canadian Arctic where the SIT is much too thin in the control (see Fig. 5).

For matchups where the BGEP SIT observations are under 1 m, the mean difference, standard deviation and RMSD between the model and in situ observations are all poorer for the SIT assimilation experiment than for the control (Figs. 15(b),(c); Table 4), despite the assimilation giving very little weight to CryoSat-2 observations under 1 m. The negative mean difference
indicates that in the Beaufort Sea region, where the BGEP moorings are located, the modelled ice in the SIT assimilation experiment is thicker than the BGEP observations. Investigation has shown that this is unrelated to positive SIT increments





| | SIT assimilation experiment | | Control | | CryoSat-2 obs | |
|---|---|---|---|---|---|---|
| For SIT observations: | < 1 m | >= 1 m | < 1 m | >= 1 m | < 1 m | >= 1 m |
| Correlation coefficient | 0.89 | 0.79 | 0.89 | 0.82 | 0.65 | 0.59 |
| Mean difference | -0.18 | -0.14 | -0.01 | 0.05 | -0.36 | -0.05 |
| Absolute mean difference | 0.24 | 0.28 | 0.12 | 0.21 | 0.38 | 0.25 |
| RMS of differences | 0.36 | 0.33 | 0.18 | 0.27 | 0.47 | 0.31 |
| Standard deviation of differences | 0.31 | 0.30 | 0.18 | 0.26 | 0.30 | 0.30 |

**Table 4.** Statistics for all matchups of BGEP SIT observations with FOAM SIT analysis and assimilated CryoSat-2 SIT observations, January 2015 - August 2017, for thickness bins less than 1 m (total matchups 35 for model and 17 for satellite observations) and equal to or above 1 m (total matchups 54 for model and 32 for satellite observations). Differences are observation minus model or satellite observation, in m. Satellite observations not available between May and September each year. CryoSat-2 observations below 1 m are given very little weight in the assimilation.

in the assimilation being spread too far from regions of thicker ice. It is likely instead to be a result of poorly-specified observation uncertainties for the assimilated CryoSat-2 data. At high latitudes there are up to around 100 observations per super-observation due to overlapping orbits of the satellite, but at lower latitudes, towards the ice edge, there can be as few as 2 observations. This means there will be an increased contribution of random error in these observations. Since the measurement uncertainty component of the OBE is based on the magnitude of the SIT observations themselves, this (along with other biases in the data) is not taken into account. Mitigating the issue of random error is an avenue for future investigation, and could include, for example, filtering the CryoSat-2 observations by latitude or by a minimum number of observations included in the super-observation. Alternatively, the Bayesian background check discussed in Sect. 2.2.2 could be used to reject observations deviating too far from the model. However, as previously discussed, this relies on the free model (without assimilation) being relatively unbiased. It is also possible that thicker modelled ice in the Canadian Arctic in the SIT assimilation experiment is being advected into this region by the model, following the general pattern of ice circulation in this region. Conversely however, the relationships between SIT and season shown on Fig. 15 illustrate that the summertime (JJA) model SIT is too thin compared to the BGEP observations. The ice mass balance budget for this area could therefore be assessed as an avenue of future investigation. Research into assimilating observations of SIT under 1 m from the SMOS (Soil Moisture and Ocean Salinity) satellite instrument is also underway, which aims to improve the representation of thinner ice in FOAM and would help to mitigate these problems.

# 6 Discussion and conclusions

The feasibility of assimilating SIT (sea ice thickness) derived from CryoSat-2 along-track measurements of sea ice freeboard into a global, coupled ocean-sea ice model, FOAM (Forecast Ocean Assimilation Model), has been demonstrated. This is a novel use of along-track SIT observations, as other centres have previously used only gridded, temporally-averaged SIT mea-

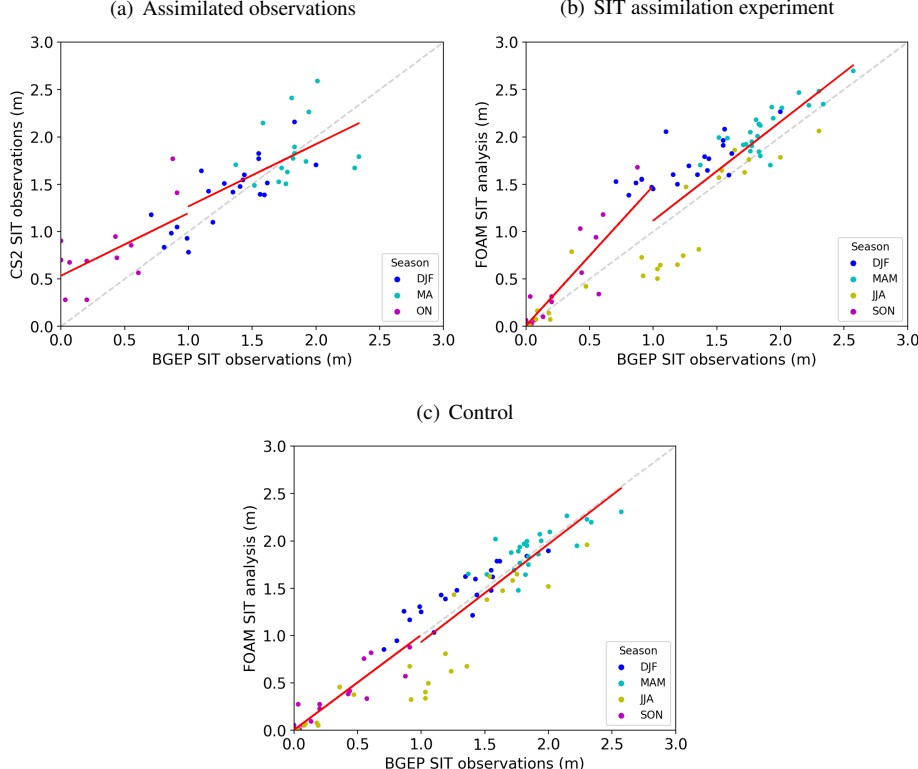

**Figure 15.** Validation of monthly gridded CryoSat-2 (CS2) observations and monthly mean FOAM SIT analyses against BGEP monthly mean observations of SIT (converted from sea ice draft) from January 2015 to August 2017, colour-coded by season. Scatter plots of BGEP observations (in m) with (a) monthly, gridded, quality-controlled CryoSat-2 SIT observations (in m); and FOAM analyses (in m) for (b) SIT assimilation experiment and (c) control. Note that CryoSat-2 matchups are not available from May to September each year). Best fit lines (in red) are plotted separately for BGEP SIT observations below 1 m, and equal to or above 1 m. 1:1 reference line shown in broken grey.

surements. The assimilation results in notable improvements to the SIT analysis and forecast fields generated by FOAM compared to a control without SIT assimilation, validated using SIT observation-minus-background mean difference and RMSD assimilation statistics. Arctic-wide improvements are 0.75 m mean difference (0.41 m RMSD) in the freeze-up period (October-505 November) and 0.46 m mean difference (0.33 m RMSD) in the ice break-up period (March-April), calculated using data from 2015-2017. Regional improvements in the Canadian Arctic are particularly notable, where there is a reduction of more than 1.30 m mean difference (1.22 m RMSD) for SIT in the break-up period.

Comparison with independent in situ SIT observations from NASA Operation IceBridge (Kurtz et al., 2019) also demonstrates that the assimilation results in substantial improvements to the model SIT, of 0.61 m mean difference and 0.42 m RMSD.
Validation against Air-EM (airborne electromagnetic induction) combined SIT and snow depth observations (Haas et al., 2009) yields poorer results. This may be evidence of noise in the SIT analysis, or uncertainty in the modelled snow depth, the assim-





ilated observations or the validation observations themselves. This may also be a result of sampling error, owing to the limited number of Air-EM matchups.

For thicknesses above 1 m, the model performance without SIT assimilation is similar to (and slightly better than) that of
CryoSat-2 SIT observations, compared to BGEP (Beaufort Gyre Exploration Project) sea ice draft measurements. Therefore, the SIT assimilation experiment does not show an improvement compared to the control against this validation dataset, although the statistics for both model runs are good. The SIT assimilation experiment experiences a more marked degradation for thicknesses below 1 m compared to the control, despite giving very little weight to SIT observations below 1 m in the analysis. The degradation may be due to poorly-specified observation uncertainties in the assimilated observations.

An unexpected slight improvement in SIT forecast performance compared to the SIT analysis may be evidence of spatial noise introduced by the assimilation, which is smoothed out over the 5-day forecast period.

Despite the lack of CryoSat-2 SIT observations over the summer months due to the presence of meltponds affecting retrievals, the model has been shown to retain improvements to the SIT field throughout the summer due to previous SIT assimilation. SIC (sea ice concentration) assessment results for the SIT assimilation experiment show a regional improvement in July
in the European sector, compared to a control. This is likely due to the slower melting of thicker ice introduced by the SIT assimilation in previous months. Overall, it is concluded that CryoSat-2 along-track measurements, rather than gridded and temporally-averaged observations, can be successfully assimilated.

The heterogeneous nature of sea ice means that SIT observations will always contain sub-gridscale noise, as well as random uncertainties in the data. The mitigating effect of using a model and a well-specified data assimilation scheme avoids the need
for gridding or temporal averaging of the noisy CryoSat-2 data. A spatial averaging of sorts is carried out on the freeboard observations through the process of super-obbing, which takes the median observation within a 10 km radius. This aims to reduce the random uncertainty in the observations, while preserving spatial variability at the model grid scale. Remaining noise in the SIT analysis could be reduced by increasing the OBE (observation error variance) for, or filtering out, super-observations with a number of input values below a specified threshold. This issue mainly affects lower latitudes close to the
ice edge, and the planned assimilation of SMOS thin ice observations (below 1 m) into FOAM will help to improve the SIT analysis and forecasts in these regions. Additionally, a Bayesian check of the observations against the model SIT assimilation background could be used, although investigation has shown the application of this method to SIT is not straightforward.

There are a number of options for further development of SIT assimilation in FOAM. Well-specified OBEs and BGEs (background error variances), and importantly the balance between them, are vital for producing high-quality analyses. The
impact of potentially unreliable thin or very thick SIT observations on the analysis is reduced by increasing their OBE, but this is currently determined from the magnitude of the SIT observations themselves and is thus reliant on their being unbiased. Measurement uncertainties generated as part of the processing chain for each CPOM freeboard observation, independent of the magnitude of the observation itself, would therefore be very useful. The specification of representation uncertainty as part of the OBE could also be refined. Owing to the reduction in Arctic sea ice cover in recent decades (e.g. Meredith et al., 2019),
a climatological BGE as used in this study may not be the optimal method for SIT assimilation. Instead, a daily-varying BGE could be obtained for SIT from the latest ensemble system under development at the Met Office. This would be combined with



the climatological BGE in a hybrid ensemble/variational framework, as is planned for the ocean variables in FOAM. Use of the NEMOVAR dual length scale capability for background error covariances could also be investigated for SIT.

Validation of FOAM modelled snow depth is planned, which is an important source of uncertainty in the conversion of sea ice freeboard to SIT. Additionally, it has been shown that snow depth in the model is sensitive to the SIT assimilation itself, and this could cause a feedback between the model and the SIT observations. It would thus be useful to conduct sensitivity studies of the modelled snow depth and apply any tuning to the model as necessary. The work in this study has also highlighted that an examination and validation of the FOAM model ice mass budget, specifically in the Beaufort Gyre region, would be useful.

In order to introduce SIT assimilation into operational Met Office forecasting systems, improvements to the delivery timeliness of the near-real-time observations would be required, as well as operational support of their dissemination. The CPOM near-real-time product used in this study currently has a latency of 72 hours and, at the present time, FOAM uses observations which are made available within 48 hours. The coupled NWP (numerical weather prediction) system being implemented in 2021 will require observations to be available as close to real-time as possible, up to a maximum delay of 24 hours.

Operational implementation of SIT assimilation in FOAM would directly influence the Met Office's GloSea seasonal forecasting system (MacLachlan et al., 2014), as this is initialised using FOAM analysis fields. GloSea requires a long timeseries of observations (ideally 25 years) of each assimilated data type to produce a reanalysis, which is used to initialise hindcasts for forecast calibration purposes. The satellite SIT observation dataset reprocessed back to 2002 by the ESA Sea Ice CCI project (Hendricks et al., 2018) and any further planned consistent reprocessing of observations from the earlier ERS-1 and ERS-2 satellites will therefore be vital for this application. Using improved FOAM SIT fields for initialisation, particularly for regions of thicker ice, is expected to be of substantial benefit to the quality of seasonal forecasts of sea ice extent, concentration and thickness produced by GloSea, as was demonstrated by Blockley and Peterson (2018).

CryoSat-2 is currently operating well beyond its 3.5-year nominal lifespan, having been launched in 2010. Freeboard observations from NASA's ICESat-2 (launched 2018, also with a design life of 3 years) are available, though not currently in near-real-time. Although planning is underway for a high-priority ESA candidate satellite mission to observe the polar regions, CRISTAL (Copernicus Polar Ice and Snow Topography Altimeter), there is a real risk of an observation data gap should CryoSat-2 and ICESat-2 both fail before the CRISTAL mission is realised. The work in this study highlights the significance of the continuation of dedicated satellite missions for monitoring SIT, and demonstrates the suitability of near-real-time SIT observations for use in operational ocean-sea ice modelling and forecasting applications.

*Author contributions.* EF set up and conducted the experiments, wrote the text and produced the figures. MM, EB and DM provided expertise on data assimilation and sea ice modelling. NF managed the Met Office contribution to the SEDNA (Safe maritime operations under extreme conditions: the Arctic case) project that provided the funding for this work. AR, AS and RT were responsible for producing and processing the CryoSat-2 sea ice freeboard observations and contributed technical expertise on their use.



*Competing interests.* The authors declare that they have no conflict of interest.

*Data availability.* CryoSat-2 along-track sea ice freeboard observations were processed by CPOM (Centre for Polar Observation and Mod-
elling) for use in this study and are available on request for non-commercial research use.

FOAM sea ice and ocean analysis and forecast products from the SIT assimilation experiment and control are available on request for non-commercial research use.

The in-situ sea ice thickness observations used for validation of the FOAM model output were obtained through the University of Washington Unified Sea Ice Thickness Climate Data Record (Schweiger, 2017). The teams responsible for the collection and processing of the
field measurements are acknowledged:

Operation IceBridge (QuickLook v1 product, downloaded 1 August 2019): Kurtz et al. (2019)

Air-EM (downloaded 1 August 2019): Haas et al. (2009)

BGEP (downloaded 5 August 2019): The data were collected and made available by the Beaufort Gyre Exploration Project based at the Woods Hole Oceanographic Institution (http://www.whoi.edu/beaufortgyre).

The SSMIS sea ice concentration observations used for validation in Sect. 4 are from EUMETSAT OSI SAF (Ocean and Sea Ice Satellite Application Facility) products OSI-401 and OSI-401-b (Tonboe et al., 2017), obtained in near-real-time through the EUMETCast delivery system.

The sea ice type data used for Fig. 1 are from EUMETSAT OSI SAF product OSI-403-c (Aaboe et al., 2020), obtained in near-real-time via ftp.

*Acknowledgements.* This work was carried out under the SEDNA project, which received funding from the European Union's Horizon 2020 Research and Innovation Programme, under grant agreement no. 723526. EB further acknowledges funding from the European Union's Horizon 2020 Research and Innovation Programme through grant agreement no. 727862 (APPLICATE). Ana Aguiar, David Ford, Jennifer Waters and James While of the Met Office are all acknowledged for providing helpful technical advice which contributed to this work.





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
