# Peer review of "Assimilation of sea ice thickness derived from CryoSat-2 along-track freeboard measurements into the Met Office's Forecast Ocean Assimilation Model (FOAM)"

_The Cryosphere, 2021_

## Referee Comment (RC2)

The paper describes assimilation of Cryosat-2 winter sea ice thicknesses (SIT) in the Met Office sea ice analysis system. The paper's main novelty is that Cryosat-2 CPOM along-track observations were used as opposed to averaged gridded and temporally-averaged products, and modelled snow thickness values were employed to convert ice freeboards to ice thicknesses as opposed to the conventionally used Warren's snow thickness climatology. Verification of the control (no SIT assimilation) and SIT assimilation experiments against various SIT observations is presented. The paper claims that there is some indication of the usefulness of assimilating Cryosat-2 winter SIT observations in sea ice numerical systems, but the results are mixed and not convincing. I believe that the paper is currently not suitable for publication and the following comments should be addressed before the paper can be considered for publication again.

General Comments:

My main concern is that the paper does not contain a clear evidence that assimilation of CPOM Cryosat-2 retrievals helps to bring modeled SIT closer to the reality. The large improvement in SIT analysis is observed when compared against Cryosat-2 biased data themselves (although not yet assimilated), and I think this improvement simply comes from the memory of the previously assimilated Cryosat-2 SIT observations. But such a result is expected regardless the quality of the assimilated Cryosat-2 observations. Thus, how are the authors confident that assimilation of Cryosat-2 observations does actually bring modeled SIT closer to the reality? Particularly, given the fact that degradation is observed when compared against ULS and EM induction data. There is some improvement with respect to Ice Bridge data, but the amount of data is limited. Overall, the paper presents mixed results which do not allow one to conclude that the currently used Cryosat-2 retrievals could be considered ready for data assimilation purposes. I believe, that much more work needs to be done to improve the quality of Cryosat-2 ice thickness retrievals before considering them appropriate for assimilation.

The authors should consider improving the Cryosat-2 freeboard retrievals by taking into account various sources of uncertainty due to various effects including:

(1) the effect of brine-wetted snow reported and quantified in (Nanden et al. 2017)
(2) the effect of surface roughness described and quantified in (Landy et al. 2020)
(3) the effect of ice type, i.e., increased Ku-band signal penetration in multi-year ice (MYI) compared to the first-year ice potentially leading to freeboard underestimation for MYI (Xia and Xie, 2018)

The authors could also consider using additional independent data sources for verification SIT analyses such as ice charts.

The authors report that the sea ice model retains improvements to the SIT field throughout the summer months, due to winter SIT assimilation. However, (Bushuk et al., 2020) investigated a so-called "Arctic spring predictability barrier", and they found that initializing sea ice models with SIT observations prior to May/June is not beneficial for

predicting summer sea ice, and, therefore, summer ice thickness observations are strongly required. The authors should discuss in the paper how their results correspond to the previous findings in (Bushuk et al., 2020, Bonan et al., 2019).

References:

Nandan, V., Geldsetzer, T., Yackel, J., Mahmud, M., Scharien, R., Howell, S., et al. (2017). Effect of snow salinity on CryoSat-2 Arctic first-year sea ice freeboard measurements. Geophysical Research Letters, 44, 10419– 10426. https://doi.org/10.1002/2017GL074506

Landy, J., Petty, A., Tsamados, M. & Stroeve, J., 2020. Sea ice roughness overlooked as a key sourceof uncertainty in CryoSat-2 ice freeboard retrievals.  J. Geophys. Res. Oceans,  125(5), p.e2019JC015820.

W. Xia and H. Xie, "Assessing three waveform retrackers on sea ice freeboard retrieval from CryoSat-2 using operation IceBridge Airborne altimetry datasets," *Remote Sens. Environ.*, vol. 204, pp. 456–471, Jan. 2018.

Bushuk, M., Winton, M., Bonan, D., Blanchard-Wrigglesworth, E., & Delworth, T. (2020). A mechanism for the Arctic sea ice spring predictability barrier. Geophys. Res. Lett., 47(13). doi:10.1029/2020GL088335

Bonan, D., Bushuk, M., & Winton, M. (2019). A spring barrier for regional predictions of summer Arctic sea ice. Geophys. Res. Lett., 46(11), 5937-5947. doi:10.1029/2019GL082947

Specific Comments:

Line 1 and throughout the text. Usually, abbreviation is given in parentheses, i.e., "SIT (sea ice thickness) → "sea ice thickness (SIT)".

Line 51. "from retrievals of brightness temperature" → "from L-band brightness temperature measurements".

Line 100. "Temperature profiles are also obtained from marine mammals." What does this actually mean?

Equation (3). Please define $e$

Tables 2, 3, and 4. Please add dimension (m) where appropriate to the first column of the table.

Line 420. "may indicate spatial noise" → "may indicate that spatial noise"

---

## Author Comment (AC1)

Response to reviewer #2

The authors thank reviewer #2 for their comments.

The review focuses on the suitability of the CryoSat-2 SIT observations for assimilation, rather than on the new methods for assimilation of this data described in the paper. Questioning the quality of the assimilated data is certainly a sensible point, but is one we are able to counter as follows. (Reviewer comments are shown below in green).

A good data assimilation scheme takes into account errors and uncertainties in the observations. Any quality of data can be assimilated, as long as the observation uncertainty is properly accounted for. This allows the analysis to give more or less weight to the observations as required. An in-depth discussion of the observation uncertainty and potential areas for improvement is already included in the paper.

*"Thus, how are the authors confident that assimilation of Cryosat-2 observations does actually bring modeled SIT closer to the reality? Particularly, given the fact that degradation is observed when compared against ULS and EM induction data. There is some improvement with respect to Ice Bridge data, but the amount of data is limited."* and *"My main concern is that the paper does not contain a clear evidence that assimilation of CPOM Cryosat-2 retrievals helps to bring modeled SIT closer to the reality."* Assessment against the independent OIB (Operation IceBridge) observations clearly demonstrates that the assimilation leads to improvements in the model output. The reviewer bases their conclusions on the Air-EM (45 matchups) and ULS (BGEP; 54 matchups for assimilated data above 1 m thickness) validation, and dismisses the conclusions from the OIB validation on the grounds that the number of matchups is limited, despite there being far more data (547 matchups). It is therefore difficult to see the reviewer's argument here.

As noted by the reviewer, the results for Air-EM matchups are worse than for OIB, but various reasons are given in the paper including that actually these, and not the OIB matchups, are limited in number (45 matchups vs 547). The ULS (BGEP) results for thicknesses above 1 m (data below 1 m are given little weight in the assimilation) are fairly similar with or without the assimilation since the validation against the CryoSat-2 data and the model are similar at this location, as discussed in the paper. Both the BGEP and Air-EM matchups have additional sources of uncertainty: conversion from draft to thickness (BGEP) and the addition of snow (Air-EM). Further discussion on this is given in the paper.

Furthermore, there are several published papers which demonstrate the good performance of the assimilated CryoSat-2 data against independent observations, e.g. Tilling et al., 2015 (already cited in the paper, but I will explicitly add this to section 5).

The reviewer suggests that we work on improving the CryoSat-2 retrievals. However, this is well beyond the scope of this paper, which is on the topic of assimilating the data that is presently available. Since there is an increasing body of peer-reviewed literature on the assimilation of CryoSat-2 observations, e.g. Yang et al. (2014); Mu et al. (2018); Xie et al. (2018); Liang et al. (2020), including in this journal, the grounds for rejecting our manuscript for doing the same would be highly questionable.

Measurement uncertainties can be accounted for in the observation uncertainty estimate used in the assimilation, and so do not necessarily need to be addressed at the retrieval stage. It is not possible currently to account for particular uncertainties in the simple thickness-based observation uncertainty estimate generated for use in this study, but it should be possible to include them in an observation uncertainty estimate generated as part of the data processing chain. Indeed, this paper provides the evidence and motivation for data producers to work on this, with the knowledge that there is a direct application for it, and users who are ready and waiting.

The Landy et al. (2020) reference for surface roughness suggested by the reviewer was already cited in the paper as part of the discussion on random uncertainty in the retrievals. However, on reflection, this is actually a systematic and not a random uncertainty (can't be removed by averaging), so actually comes under the measurement uncertainty (section 2.4.1). Similarly, the Nandan et al. (2017) work on ice salinity would be covered by this too. We will add more information to explicitly list examples of the sources of error that are covered by the measurement uncertainty.

The effect of sea ice type is already accounted for at the retrieval stage (e.g. Tilling et al., 2018).

The stated aim of this paper is to demonstrate that the assimilation of along-track CryoSat-2 observations is feasible, i.e. a "proof of concept". The further work that needs to be done before the assimilation can be included in an operational system is fully acknowledged in the paper. An in-depth discussion of the weaknesses in the current observation uncertainty estimate is already included in the manuscript, but this does not mean that the dataset has no value for assimilation at all. It is clear that the modelled sea ice is much too thin; the assimilated data therefore does not have to be of high quality to contribute any improvement. Additionally, there is certainly the expectation that, like all the satellite observations currently assimilated in FOAM, CryoSat-2 freeboard retrievals will continue to undergo further improvements and corrections by the data producers. It is not necessary to wait until the data is of extremely high quality before attempting to assimilate it.

We have therefore shown that, in forming their conclusion that assimilating winter SIT observations in sea ice numerical systems is not useful, the reviewer has not duly considered the impact of the data assimilation system on the analysis in addition to the observations. They have also not given proper consideration to the discussion already included in the paper on observation uncertainties. They have additionally discounted the strong evidence of improvement in the model performance validated using independent OIB observations, in favour of datasets with far fewer matchups, and have not taken into account the in-depth discussion in the paper of why these latter results should be considered less reliable. They have additionally not considered previously published validation of the CPOM CryoSat-2 observations.

It remains unclear why, even if the SIT improvements due to the assimilation were marginal (or even demonstrated a clear detriment), this would be grounds for rejection of the paper as that result would still be useful to know. Presumably the methodology and assessment in this study are sound, since the reviewer has made no comment on those. No suggestions have been offered on how to improve the assimilation, other than to upgrade the dataset prior to use, which is well beyond the scope of the project.

Addressing specific points in the review:

The reviewer states *"The large improvement in SIT analysis is observed when compared against Cryosat-2 biased data themselves (although not yet assimilated), and I think this improvement simply comes from the memory of the previously assimilated Cryosat-2 SIT observations. But such a result is expected regardless the quality of the assimilated Cryosat-2 observations."* This is correct. Assessment of observation-minus-background statistics is standard in data assimilation, and this demonstrates that the assimilation is working as expected. The paper does not claim that the results demonstrate a reduction in model bias, see discussion of e.g. reductions in "mean difference" and not "mean error". What it does show is that the model is brought into line with the observations, which have been independently verified (e.g. Tilling et al., 2018).

*"The authors could also consider using additional independent data sources for verification SIT analyses such as ice charts."* Ice charts show sea ice concentration, rather than thickness. However, in a follow-on paper, further validation of SIT assimilation in the FOAM system, including the effects on the modelled sea ice concentration, will be shown.

*"The authors report that the sea ice model retains improvements to the SIT field throughout the summer months, due to winter SIT assimilation. However, (Bushuk et al., 2020) investigated a so-called "Arctic spring predictability barrier", and they found that initializing sea ice models with SIT observations prior to May/June is not beneficial for predicting summer sea ice, and, therefore, summer ice thickness observations are strongly required. The authors should discuss in the paper how their results correspond to the previous findings in (Bushuk et al., 2020, Bonan et al., 2019)."*

The reviewer is referring to seasonal forecasting, whereas this is not what our model is doing over the summer. The system continues to run over the summer months, assimilating all other available data types daily, and using them to initialise only short-term (5-day) forecasts. It is agreed that summer ice thickness observations would be great, if only they were available! In any case, e.g. Blanchard-Wrigglesworth and Bitz (2014) found sea ice thickness anomalies in general circulation models to have a timescale of between 6 and 20 months, so seeing the impact of spring SIT assimilation in September is not unexpected. Seasonal forecasting is briefly discussed in the paper already, but further discussion as suggested would be beyond the scope of the paper.

Specific Comments:

*Line 1 and throughout the text. Usually, abbreviation is given in parentheses, i.e., "SIT*

*(sea ice thickness) - "sea ice thickness (SIT)".* Changed

*Line 51. "from retrievals of brightness temperature" - "from L-band brightness*

*temperature measurements".* Changed

*Line 100. "Temperature profiles are also obtained from marine mammals." What does*

*this actually mean?* Changed text to "Temperature profiles are also obtained from instrumented marine mammals (Carse et al., 2015)."

*Equation (3). Please define e* e is the mathematical constant (2.71828). Text updated to make this clear.

*Tables 2, 3, and 4. Please add dimension (m) where appropriate to the first column of*

*the table.* Added

*Line 420. "may indicate spatial noise" - "may indicate that spatial noise"* Changed

Additional, as mentioned above:

Moved Landy et al. (2020) reference, as incorrect in the context of random uncertainty, and added to discussion of measurement uncertainties along with Nandan et al. (2017).

Added Tilling et al. (2015) reference to section 5 to illustrate independent validation of CryoSat-2 observations.

---

## Author Comment (AC2)

Response to Reviewer #1 comments

The authors thank Reviewer #1 for their comments. Responses to specific comments are below (reviewer responses in blue).

*Snow thickness is difficult to simulate in sea ice models. The unreliable precipitation and the lack of robust observations could be attributed to as possible reasons. However, since the conversion from the freeboard to thickness relies majorly on how well the snow thickness is simulated, I wonder if the authors could provide some sentences to discuss the sensitivity of the forecast results to the simulated snow thickness.*

Added further discussion, from line 147 now reads:

Currently, CPOM makes use of a modified snow depth climatology, based on Warren et al. (1999) and halved over first-year ice, for processing CryoSat-2 sea ice freeboard retrievals and conversion to SIT (Tilling et al., 2015). This approach is also used by other centres processing CryoSat-2 freeboard observations: Alfred Wegener Institute (AWI; Ricker et al., 2014) and NASA (Kwok and Cunningham, 2015). Instead, here the FOAM modelled snow depth is used. Modelled snow depth has a greater spatial and temporal variability than can be obtained from a climatology, as demonstrated by Mallett et al. (2021) and illustrated on Fig. 1. Using this method also maintains consistency between SIT and snow depth within the FOAM model. A preliminary validation indicates that the FOAM snow depth is somewhat thinner than the modified climatology of Warren et al. (1999), as shown on Fig. 1, particularly over multi-year ice. Tuning experiments demonstrate that simply increasing the snow depth in the model does not result in better evaluation of the SIT analysis against independent observations, owing to feedbacks in the model and between the SIT assimilation and the snow depth itself.

Snow depth uncertainty is a large source of error in radar altimetry sea ice measurements, both in the retrievals of freeboard and the subsequent conversion to SIT (e.g. Giles et al., 2007; Ricker et al., 2015). Due to the linear relationship between SIT and snow depth (Eqns. 1 and 2), an underestimation of the snow depth would lead to an underestimate in the SIT. Large uncertainties in the snow depth may apply whether it has been modelled or taken from climatology. Additional uncertainty is also introduced in Eqn. 1 through lack of knowledge of the snow and sea ice densities which, although constants in the CICE model used here, are spatially and temporally varying in reality (e.g. Alexandrov et al., 2010; Kern et al., 2015). Uncertainties due to variations in water density can be neglected (Ricker et al., 2014; Kurtz et al., 2014). In order to quantify and reduce the uncertainty in the FOAM modelled snow depth, future plans will include the assimilation of satellite snow depth observations.

Reference: Mallett et al., (2021): The Cryosphere, 15, 2429–2450. https://doi.org/10.5194/tc-15-2429-2021

Other references as already cited in the paper.

*L142: make -> makes* Changed

*Eq (3): The threshold for thickness is 0.7m when having an uncertainty of 8. I realize that the authors have explained how the shape of these functions are obtained, but I feel curious of why 0.7m is used. If it was arbitrarily selected, then this information is necessarily to be present in the context.* 8 m was the arbitrarily selected value as the maximum uncertainty for thin ice observations, being a very large value compared to the observations themselves. 0.7 m is simply where the function happens to reach 8 m. This has been clarified in the text:

"8 m is an arbitrarily-selected value of the maximum uncertainty for thin ice observations, being a very large value compared to the SIT observations themselves. This ensures that these data do not influence the analysis. 0.7 m is where the function reaches this value."

*Paragraph L370: About the unexpected improvement in the mean differences. The authors state that it is caused by the spatial noise introduced during data assimilation. That could be one of the reasons, but from my side, I tend to believe it is caused by the systematic errors of the model. In Figure 5b,c, the model shows negative thickness bias, which indicates a slower growth during the freezing period. However, as suggested in Figure 9a, slightly thicker ice than observations are generally found. That is to say, the assimilation introduced thickness increment is faster than the ice growth by the model physics, i.e., ice grows faster in observations than in models. Could that be the case? I have no evidence about that.*

Yes, this could also be an explanation. Have added to line 373: "This may also be a result of observed ice growing faster than in the model, which has a systematic error towards thinner ice in the control (figure 5(b); figure 11(b))."

*The discussion and conclusions could be made compact. I currently read it feeling too much redundant information. I would suggest one or two future plans are already enough for wrap-up.*
Changes have been made to cut down this section. However, since this the aim of this paper is to demonstrate that the assimilation is feasible, and emphasises that the work is not complete, it is useful to summarise the future work that is required. It is also important to communicate to SIT observation data producers what is needed from them in order to move forwards.

---

## Author Comment (AC3)

Responses to Reviewer #3 comments

The authors thank Reviewer #3 for their comments.

Responses to the specific comments are below, with reviewer comments in brown.

*Specific Comments:*

*Line 105: What is horizontal resolution of the atmospheric forcing? Please include this in the text.* Added (17 km).

*Page 14 Fig 4: add 87.5 N gridline on Fig. 4a and 4b to make the area more discernable.* Done

*Line 357-358: "Improvement" in Atlantic sector is (5%) is small and difficult to see in plot. Can you make a difference plot between 10c and 10d to show this?* The changes are also seen in the mean difference plots (10a,b) not just the RMSD plots (10c,d). In order to avoid adding two extra plots, and because it seems the previous wording in the paper was too vague, instead on line 357 we have added "specifically in the area immediately north of Svalbard and Franz Josef Land" after "in the Atlantic sector", to clarify where to look.

*To augment the NASA IceBridge, BGEP ULS moorings and Air-EM observations, the paper would be more comprehensive if Dartmouth/CRREL IMB were included. Specifically 2015D and 2015F (http:// http://imb-crrel-dartmouth.org/archived-data/). These buoys are outside the Beaufort Sea region, where little improvement was found with the assimilation of CryoSat-2 along-track freeboard data. How do these observations compare with the FOAM ice and snow depths? I would like this analysis added to the paper.*

An assessment of model and CryoSat-2 observations against the Dartmouth-CRREL Ice Mass Balance buoy dataset was conducted in response to this comment. However, the results were suspicious. Unfortunately, there are concerns with the both the spatial representativeness and the quality of the IMB SIT dataset, making it unsuitable for use as a validation reference in this paper.

Firstly, producing meaningful matchups of the mean gridbox ice thickness of the model with the IMB dataset is difficult. The model is attempting to simulate the average behaviour of the ice field as an ITD (ice thickness distribution), whereas the IMB observations are point measurements moving with discrete ice floes. Sampling bias in these point measurements has been previously demonstrated in the literature, where IMB observations have been unable to capture the full range of ice thicknesses present (e.g. West et al., 2020). Consequently, the IMB measurements cannot be assumed to be representative of the wider region, and the average behaviour of the ice as represented by the model.

Additionally, it has also been demonstrated that there are issues with the quality of IMB snow depth measurements (e.g. Blanchard-Wrigglesworth et al., 2018). A poor snow depth observation would result in an unreliable SIT observation. If necessary, we are also able to provide supplementary

material showing further assessment of a large number of IMB snow depths against modelled and observed snow depth climatologies, which draws similar conclusions. The snow depths for the specific buoys 2015D and 2015F as suggested by the reviewer, and also 2015G and 2015J, were examined in response to the reviewer's comment and these also showed similar issues. Unfortunately, it must be concluded that the IMB SIT observations are not sufficiently reliable to be used as a reference for validation in this paper.

References:

West, A., Collins, M., and Blockley, E. (2020). Using Arctic ice mass balance buoys for evaluation of modelled ice energy fluxes, Geosci. Model Dev., 13, 4845–4868, https://doi.org/10.5194/gmd-13-4845-2020

Blanchard-Wrigglesworth, E., Webster, M. A., Farrell, S. L., and Bitz, C. M. (2018). Reconstruction of snow on Arctic Sea Ice, Journal of Geophysical Research: Oceans, 123, 3588–3602, https://doi.org/10.1002/2017JC013364

*The paper does not show any comparison to IABP ice drift data. Although not a fully coupled air-ocean-ice modeling system, have you examined ice drift and if you have, did you see any improvement in ice drift prediction between the Control and CryoSat-2 assimilative hindcast? If you have not , what is the level of effort to incorporate an ice drift analysis to the existing manuscript?* A quick comparison of FOAM ice drift fields shows that there is some difference between the SIT assimilation run and the control. However, this is smaller than the difference between either model run compared with NSIDC v4 sea ice motion vectors (which also include IABP buoys). We have not compared FOAM to in situ ice drift observations directly. Since ice drift has not been extensively validated in FOAM, it would be a substantial amount of work to include a proper assessment in what is already quite a long paper. However, it is a useful suggestion and will hopefully be included in future work plans.

---

## Author Comment (AC4)

Responses to Reviewer #4 comments

The authors thank Reviewer #4 for their comments. Responses to the comments are given below, with reviewer comments shown in red.

*General Comments*

*If I understand correctly, the modelled SIT using assimilation of CS2, and the control run (no assimilation) are compared with CS2 SIT observations that are also used for assimilation? But then it seems somehow clear that the modelled SIT using assimilation of CS2 performs better. Perhaps it would be better to evaluate with a more independent product, e.g. CS2SMOS (at least this is a different product).*

Assessment of observation-minus-background statistics is standard in data assimilation, and demonstrates that the assimilation is working as expected. It is also a form of validation, since the model is brought into line with the CS2 observations and these have been independently verified (e.g. Tilling et al., 2018). It is agreed that comparison to independent data is also important, which is why the SIT analyses and forecasts are subsequently compared to various independent in situ datasets.

Note that the FOAM modelled SIT will be assessed against the CS2SMOS product in a follow-on paper on the assimilation of CS2 and SMOS together, and therefore it is not necessary to include it in this paper.

*I also assume that the modelled snow depth is a very important component here and potentially adds significant uncertainty. A detailed evaluation of the modelled snow depth within FOAM would be important as well for evaluation here, but I understand if this is not within the scope of this paper, and the authors also suggest carrying out such a study in future.*

It is agreed that the modelled snow depth is a very important component of the conversion of freeboard observations to SIT. Further discussion of this has been added to the paper. Future plans also include the assimilation of satellite snow depth observations in FOAM. A more comprehensive evaluation of the modelled snow depth will take place as part of that work, and is thus beyond the scope of this paper.

From line 147 now reads:

Currently, CPOM makes use of a modified snow depth climatology, based on Warren et al. (1999) and halved over first-year ice, for processing CryoSat-2 sea ice freeboard retrievals and conversion to SIT (Tilling et al., 2015). This approach is also used by other centres processing CryoSat-2 freeboard observations: Alfred Wegener Institute (AWI; Ricker et al., 2014) and NASA (Kwok and Cunningham, 2015). Instead, here the FOAM modelled snow depth is used. Modelled snow depth has a greater spatial and temporal variability than can be obtained from a climatology, as demonstrated by Mallett et al. (2021) and illustrated on Fig. 1. Using this method also maintains consistency between SIT and snow depth within the FOAM model. A preliminary validation indicates that the FOAM snow depth is somewhat thinner than the modified climatology of Warren et al. (1999), as shown on Fig. 1, particularly over multi-year ice. Tuning experiments demonstrate that simply increasing the snow depth in the model does not result in better evaluation of the SIT analysis against independent observations, owing to feedbacks in the model and between the SIT assimilation and the snow depth itself.

Snow depth uncertainty is a large source of error in radar altimetry sea ice measurements, both in the retrievals of freeboard and the subsequent conversion to SIT (e.g. Giles et al., 2007; Ricker et al., 2015). Due to the linear relationship between SIT and snow depth (Eqns. 1 and 2), an underestimation of the snow depth would lead to an underestimate in the SIT. Large uncertainties in the snow depth may apply whether it has been modelled or taken from climatology. Additional uncertainty is also introduced in Eqn. 1 through lack of knowledge of the snow and sea ice densities which, although constants in the CICE model used here, are spatially and temporally varying in reality (e.g. Alexandrov et al., 2010; Kern et al., 2015). Uncertainties due to variations in water density can be neglected (Ricker et al., 2014; Kurtz et al., 2014). In order to quantify and reduce the uncertainty in the FOAM modelled snow depth, future plans will include the assimilation of satellite snow depth observations.

Reference: Mallett et al., (2021): The Cryosphere, 15, 2429–2450. https://doi.org/10.5194/tc-15-2429-2021

Other references as already cited in the paper.

*In fact, looking at the validation results using independent SIT measurements, the improvement from the assimilation is not so obvious anymore. Moreover, looking at Fig 7 (at the very beginning of each autumn), it seems that performance significantly decreases through the summer for both the assimilation and control run.*

Once the daily SIT assimilation is stopped at the start of each summer, the model performance would not be expected to continue in the same way. Assimilation can't "fix" model biases, which return when the assimilation ceases. However, the results show that even after several months the model does not drift entirely back to its pre-assimilation state. This is not really a major part of the evaluation, but is included as an interesting point, demonstrating that there is some memory of the assimilation retained over the summer when SIT observations are not available for assimilation.

*I think this discrepancy in the evaluation needs to be discussed in more detail…* As above, a drift back to the model baseline in the absence of observations would be expected of any assimilation system.

*…so it is clearer to the reader how good this assimilation really works and if it is a benefit using along track data instead of gridded products for assimilation. A comparison using gridded products for assimilation would be beneficial as well. Is it really an advantage to use along-track measurements? This should be discussed in more detail.*

At this stage it is not of clear benefit to assimilate the along-track product over a gridded product, since the observation uncertainty for the along-track observations needs more work, as discussed in the paper. However, the main advantages of along-track data over gridded data are the reduction in spatially-correlated uncertainties and the determination of more accurate uncertainty estimates than is possible for data with more processing applied (this is already mentioned in the paper). As also discussed, developments towards coupled ocean-ice-atmosphere NWP at the Met Office require short assimilation time windows of the order of 6 hours, which means that along-track rather than temporally-averaged observations are required.

The paper demonstrates that the assimilation of along-track SIT data is feasible, i.e. that using the data in a different way than has been done previously is possible. This is the first step towards

getting the things we need in place, and provides evidence for data producers that users are ready and waiting for along-track SIT data produced in near-real-time, and emphasises the importance of well-defined observation uncertainty estimates produced as part of the processing chain. When this is all available, the assimilation of along-track SIT data will lead to demonstrable improvements over the gridded SIT products for our systems.

*Specific Comments:*

*L 114: It would be interesting to SIT maps also for the summer period. A figure showing SIT maps with and without assimilation throughout one season would be beneficial.*

Figure 6 shows SIT difference plots for the SIT assimilation experiment minus the control, for the start and end of the 2016 summer period. Although a full set of SIT maps for the experiment and control runs would give more detail, the conclusions drawn would be the same as already included in the text. Additionally, there are already a large number of figures in the paper. Therefore, no changes have been made.

*L 152-153: probably also mention dual altimetry, i.e., Ku/Ka band, e.g., Lawrence et al.* Have taken this sentence out as future plans are now to assimilate snow depth observations in FOAM (which has been added to the paper here).

*L 170: Why median and not mean?* Using the median prevents outliers from influencing the result (have added this to paper).

*L206: These are not helicopter measurements. PAM-ARCMIP measurements were carried out with a fixed wing aircraft.* Replaced "helicopter" with "aircraft".

*L 339-347 and Figure 7: But there is a quite high mean difference/RMS at the beginning of autumn when CS2 data for assimilation become available again. But that means that the performance decreases significantly within summer months also for the model using assimilation, see also my general comments?* Yes. This is expected, see responses to comments above. Once the assimilation stops, the model begins to drift back to its baseline state.

*Figures: I recommend having all relevant information included in the figure. Sometimes this information must be searched for in the caption, which slows down the flow, e.g., Figure 6: colorbar label, Figure 7, 11: dashed lines. I think it is easy to provide that information in the legend.*

Added information to legend Figs. 7, 11. However, unclear what to put in a colorbar label for Figure 6? SIT difference (m) is shown in title, and is in the same format as the other figures. We will change though if it doesn't follow journal guidelines.

---

## Author Comment (AC5)

Response to Reviewer #5 comments

The authors thank Reviewer #5 for their comments. Responses to specific comments are below, with reviewer comments shown in purple.

1) In the assimilation experiment, the converting from the freeboard Fi to sea ice thickness is a

key part to affect the SIT forecast performance in FOAM. Compared with the common

converting tool like Round Robin Data Package (ESA, 2013), the authors use the model snow

depth to replace the climatology. If possible, showing these two types of along-track SIT could

be interesting and meaningful to understand the large contrast between the different SIT

observation products in Fig. 12c and Fig. 13c.

It is acknowledged that snow depth is an important source of error in the conversion from freeboard to SIT. However, modelled snow depth has a greater spatial and temporal variability than can be obtained from a climatology, as demonstrated by Mallett et al. (2021) and illustrated on Fig. 1. Using this method also maintains consistency between SIT and snow depth within the FOAM model. It should be noted that large uncertainties in the snow depth may apply whether it has been modelled or taken from climatology.

Figure 12(c) shows the good relationship between CryoSat-2 SIT produced using the modelled snow depth and Operation IceBridge SIT observations, and demonstrates that there is no reason to believe that the modelled snow depth is poor, or adversely affecting the SIT observations. Uncertainty in the snow depth will be contributing to differences between Figs. 12(c) and 13(c), although the difference between the CryoSat-2 and Air-EM observations is greater than the snow depth itself and is therefore not the main source of this difference.

In order to quantify and reduce the uncertainty in the FOAM modelled snow depth, future plans will include the assimilation of satellite snow depth observations. It is agreed that a comparison of the effect of modelled and climatological snow depth on freeboard conversion to SIT would be interesting, and it should be included as part of that work. An in-depth comparison is therefore beyond the scope of the current paper, whose main focus is to demonstrate the assimilation of the along-track observations.

For information, discussion on this topic in the paper has been expanded, from line 147:

Currently, CPOM makes use of a modified snow depth climatology, based on Warren et al. (1999) and halved over first-year ice, for processing CryoSat-2 sea ice freeboard retrievals and conversion to SIT (Tilling et al., 2015). This approach is also used by other centres processing CryoSat-2 freeboard observations: Alfred Wegener Institute (AWI; Ricker et al., 2014) and NASA (Kwok and Cunningham, 2015). Instead, here the FOAM modelled snow depth is used. Modelled snow depth has a greater

spatial and temporal variability than can be obtained from a climatology, as demonstrated by Mallett et al. (2021) and illustrated on Fig. 1. Using this method also maintains consistency between SIT and snow depth within the FOAM model. A preliminary validation indicates that the FOAM snow depth is somewhat thinner than the modified climatology of Warren et al. (1999), as shown on Fig. 1, particularly over multi-year ice. Tuning experiments demonstrate that simply increasing the snow depth in the model does not result in better evaluation of the SIT analysis against independent observations, owing to feedbacks in the model and between the SIT assimilation and the snow depth itself.

Snow depth uncertainty is a large source of error in radar altimetry sea ice measurements, both in the retrievals of freeboard and the subsequent conversion to SIT (e.g. Giles et al., 2007; Ricker et al., 2015). Due to the linear relationship between SIT and snow depth (Eqns. 1 and 2), an underestimation of the snow depth would lead to an underestimate in the SIT. Large uncertainties in the snow depth may apply whether it has been modelled or taken from climatology. Additional uncertainty is also introduced in Eqn. 1 through lack of knowledge of the snow and sea ice densities which, although constants in the CICE model used here, are spatially and temporally varying in reality (e.g. Alexandrov et al., 2010; Kern et al., 2015). Uncertainties due to variations in water density can be neglected (Ricker et al., 2014; Kurtz et al., 2014). In order to quantify and reduce the uncertainty in the FOAM modelled snow depth, future plans will include the assimilation of satellite snow depth observations.

Reference: Mallett et al., (2021): The Cryosphere, 15, 2429–2450. https://doi.org/10.5194/tc-15-2429-2021

Other references as already cited in the paper.

2) As Line 286 of "an estimate of 50 km for theminimum SIT correlation length scale." It can

infer the spatial scale for assimilation of SIT used 50 km. If that is true, it may be one of reasons

why the increments shown in Fig. 14 (a) and (b) are still noise. So there are two related

comments: 1) How about to compare the used horizontal scale for the SIC assimilation in this

system? 2) Have you tried to increase this scale from 50 to 100 km which is a practical order of

the spatial scale for averaging RA data.

Yes, 50 km was used, see line 287: "This value was used as a constant length scale everywhere, except...".

1) The horizontal scale for SIC assimilation in FOAM is 25 km. Preliminary testing was done using 25 km for SIT, but this resulted in spatial noise and defined track lines appearing in the SIT analysis field, indicating it was too short. As described in the paper, subsequent calculations for SIT put the minimum length scale at 50 km, which improved the analysis results.

2) Figure 4 shows that over the centre of the ice pack, where orbit track lines overlap, the SIT increments do not have gaps between them (unlike for 25 km, not shown in the paper). Choosing the length scale is a balance between not having distinct orbit tracks, but not "blurring" (smoothing) the small scale features captured by the observations. The SIT increments don't look especially noisier than the increments from other observation types assimilated in FOAM (not shown). We haven't tested the SIT assimilation using a 100 km length scale throughout (aside from in the pole hole). However, 100 km is much larger than the grid scale (~10 km in the Arctic) and this may smooth out small-scale features in the observations. However, it may well improve issues of noise in the analysis. Further refinement of the appropriate length scale for SIT (and potentially including a dual correlation length scale, as mentioned in the paper) will be carried out as part of future work. It is however beyond the scope of this initial study, which has used a recognised method (the "Canadian Quick") in order to make the first calculations.

3) To convert the sea ice draft from BGEP by dividing observations by 0.89 is not good for the validations in the ice brake-up and freeze-up months, due to it omits the snow existing. The big issue is that it will mix with the system bias of SIT and return to be detrimental of the analysis in Fig. 15. So I suggest to investigate the best fit lines in the scatterplot are divided into the three interesting periods by the months like MA, ON, MJJAS. It will be helpful to shed light on the MA evaluation to contrast with the validation in Fig. 13 in Beaufort Sea.

It is quite difficult to interpret four best fit lines (including DJF too) on one plot, and the number of observations for the two-month groupings (over all years) is too small to produce statistically significant results. Instead, we have retained the two best fit lines for above and below 1 m on Fig. 15 (which illustrate important issues in the assimilation, as described in the paper) but regrouped the coloured plotting to show MA, MJJAS, ON, DJF.

The BGEP validation in MA 2015-2017 is fairly similar for the SIT assimilation experiment and control (Fig. 15), and this is also the case for Air-EM matchups for April 2015 in the Beaufort Sea (Fig. 13). However, in ON there is an issue with ice being too thick in the SIT assimilation experiment compared to BGEP observations (Fig.15), and a comment on this has been added to the text, relating it to the poor performance for thicknesses under 1 m.

Also added to end of line 222: "…although note this does not take into account the presence of snow on the surface of the ice, which will be the case outside of the summer months." This would also presumably be detrimental to the assessment of the control as well as the SIT assimilation experiment.

4) In the section 2.4, the assimilation of SIT uses IAU as well, and the SIT increments how to

feedback on the 5 categories ice in sub-grid, although there are some words about "in

proportion to the initial volume distribution". It should be paid more words or show one

example for the reader to well understand how they can work together.

Text updated, from line 233:

"Following Blockley and Peterson (2018), SIT increments are added to each of the five sub-grid SIT categories, if the ice concentration within that category is above 1 %. The initial fraction of the contribution of each category to the gridbox mean ice volume is calculated, and that fraction of the SIT increment is then added to that category. This maintains the initial volume distribution of ice (and ice area) across each sub-grid SIT category."

Other general comments:

1) Line 97 at P4: "No SIT observations are currently assimilatedoperationally". It is better to

use "No SIT observations are currently assimilatedoperationally in the system." Changed (to "in FOAM" rather than "in the system" for clarity.)

2) Line 193 at P7: "..for the SIT assimilation experiment period at the time of assessment."

Replaced by "… for assessment of the SIT assimilation in the experimental period." Not changed, as what is meant is that the V2 dataset for 2015-2017 wasn't available when the work was being done (but may subsequently become available). This has a different meaning to the suggested replacement wording.

3) Line 246 at P10: the representation uncertainty is set to 0.05 m. Does it mean the

minimal observation error is about 0.1 m as the minimal value around the 3m SIT shown

by the curve in Fig. 3a? If right, it could be better to be presented on this panel. And this

setting could be too small, compared with previous studies and other observation

platforms. Yes, that is correct - the measurement uncertainty and representation uncertainty are combined to produce the total OBE (observation error variance). However, we would prefer to keep it separate on Figure 3, to clearly illustrate the measurement uncertainty parameterisation. Figures 3(a,b) state in their titles that they are showing measurement uncertainty. However, the figure caption has been corrected, replacing "Observation uncertainty estimates for SIT." with "Observation measurement uncertainty estimates for SIT."

An initial sensitivity study was conducted to find the minimum SIT measurement uncertainty and the shape of the parameterisation curve which worked best with the assimilation system, that is, the optimum balance between the OBE and BGE (model background error covariance). The resulting measurement uncertainty curve is shown in Figure 3(a). However, the reviewer is correct that the minimum measurement uncertainty is rather small, but this is probably achieving a good balance since the BGE is also likely underestimated, as mentioned in the paper. The model BGE will need refining in the future, particularly if (most likely larger) measurement uncertainty estimates are produced alongside the observations. Have changed text from line 289:

"Sensitivity tests were conducted to produce the optimum SIT analysis by finding an appropriate balance between the OBE and the model background error variance (BGE; Sect. 2.4.2). This allowed the final form of the OBE function to be tuned and the minimum uncertainty assigned to the most reliable observations to be set. The minimum OBE is rather small compared to previous studies (e.g. Tilling et al. (2018) gives the accuracy of CryoSat-2 SIT as 13 cm) and this likely due to an underestimate in the model BGE (Sect. 2.4.2)."

4) Line 353 at P18: "The largest regional improvements are in March-April, where the

mean difference is above 1.30 m and 1.22 m for the RMSD." Is it possible to specify

where the regions are involved. Changed to "The largest regional improvements are seen in March-April (Fig. 8), with reductions in the mean difference of more than 1.30 m and 1.22 m in the RMSD, in the European sector and parts of the Canadian Arctic."

5) Line 396 at P 23, the 30-day periods were chose to cover the observation days. As

shown in Table 1 of Section 2.3.1, the observations are located only in one or two days

in the April so the 30-day window could be too wide and far away from the reality

condition.

Calendar months were only used when producing matchups with the BGEP dataset (as stated on line 397). When producing the matchups with the OIB and Air-EM data, a 30-day period centred on the middle of the observation period was selected for averaging the CryoSat-2 data. The exception to this was OIB data for 2016, which was only available for 20 and 28th April. Since 30 days of CryoSat-2 observations are required, and these cease at the end of April each year, the 30 days was taken from 1-30th April 2016. However, this does not seem to have had a detrimental impact on results. The text has been updated to make this clear (from line 396):

"The 30-day periods were chosen to centre on the middle of the observation window of each yearly field campaign for the OIB and Air-EM validation datasets, and calendar months were used when producing matchups with the BGEP dataset. The necessary exception to this was for OIB in 2016,

with fieldwork dates of 20, 28 April (Table 1). Here, the period of 1-30 April was used in order to acquire 30 days of CryoSat-2 data, since no observations are available after 30 April each year until production resumes in the autumn. This means that OIB matchups with CryoSat-2 in 2016 may not be representative of the true relationship between the datasets. Nevertheless, these matchups are not outliers of the OIB matchup group shown on Fig. 12(c), indicating a comparable level of accuracy."

6) Fig.13: The observations from Air-EM are clearly located into two regions: Beaufort Sea

and north of Canadian. The related scatterplot separated into these two regions may be

more helpful on physic to find something and compared with the result in Fig. 15.

Results for the Beaufort Sea have now been plotted as squares on Figs. 13(c-e), and the Canadian Arctic has been left as filled circles. However, since there are only 13 matchups in the Beaufort Sea from this dataset, the statistics have been left all together in Table 3 to avoid issues with statistical significance. There is already a discussion in the text separating the results for the Canadian Arctic and the Beaufort Sea, but the addition of the location data on the scatterplots backs up these points and so references to Fig. 13 have been updated in the text to refer to these subplots too.

On the comparison to Fig. 15, we have added the following to the end of Sect. 5.3 (BGEP assessment):

"The BGEP data for March-April 2015-2017 (Fig. 15a) compares better with the CryoSat-2 observations than the Air-EM data in the Beaufort Sea for April 2015 (Fig. 13a). This indicates that the Air-EM data may be less reliable, as discussed in Sect. 5.2."

---

## Referee Report (RR1)

I have read through the latest version of the paper "Assimilation of sea ice thickness derived for CryoSat-2 along-track freeboard measurements into the Met Office's Forecast Ocean Assimilation Model (FOAM)". I feel that the authors have addressed all reviewer's comments and concerns. The revised paper represents a significant development for the ice modeling community by showing how the CPOM data can be assimilated into a sea ice modeling system during the non-summer months and provide a pathway for future enhancements using freeboard data from IceSat-2, especially if data latency issues can be resolved. As a science community, we should all strive to share Arctic ice thickness or freeboard data to allow a more comprehensive evaluation of such systems in the future.

---

## Author Response (AR2)

*Reviewer comments from Report #2*

*The OIB ice thickness data set is derived from the ATM data (snow+ice surface elevations from laser altimetry). So, ice thickness is derived indirectly from sea ice + snow surface elevations, sea surface height estimation, and snow depth derived from the snow radar. And the conversion from snow freeboard to ice thickness multiplies the uncertainties in freeboard retrievals. In contrast, Air EM directly measures sea ice + snow thickness. Therefore, I doubt that OIB measurements are generally more reliable than Air EM or BGEP draft. Of course, also Air EM comes with uncertainties, and so do BGEP and OIB. And it could be that there is a problem with the AIR EM data here rather than with the OIB, but we don't know. I think we have to live with the fact that there are differences in the validation data sets due to the different retrieval methods, impact of snow, different surface types, surface roughness, footprints, etc.*

*So, what I am criticizing is that in the paper (and the mentioned line in the response letter), it comes across that OIB is "more reliable" because it fits better to the CryoSat-2 observations (and the model). But I don't think that this is a valid conclusion. Another CryoSat-2 product might fit better to the Air EM and BGEP data. So, I suggest reconsidering these statements (see above). I don't think it will change the main conclusions of the paper.*

Author response:

Many thanks to the reviewer for their additional comments, and their useful explanation of the derivation of the independent validation datasets used in this paper. Reference to the "reliability" of the Air-EM data has been removed from the sections of the paper highlighted by the reviewer (and the abstract).

It should still be noted and discussed in the paper that the model and the CryoSat-2 observations fit better to the OIB data (and the BGEP data) than they do to the Air-EM data. Since there was unfortunately only one year's worth of Air-EM observations that overlapped with the time period covered in this study, this yielded only 45 matchups with the model for Air-EM, compared to 547 for OIB. Therefore, it seems possible that, even considering the Air-EM dataset to be extremely reliable, there is likely to be an element of sampling uncertainty affecting the results.

Therefore, we agree with the reviewer's criticism that the results do not indicate that the Air-EM observations are unreliable, and have taken this out, but have left in the discussion of sampling uncertainty potentially being the reason for the poorer results compared to OIB.

Changes made:

**Lines 16-18:**

*"This may be evidence of uncertainty in the Air-EM validation observations, sampling error, noise in the SIT analysis, or uncertainties in the modelled snow depth or the assimilated SIT observations."*

Changed to:

"This may be evidence of sampling uncertainty in the matchups with the Air-EM validation dataset, owing to the limited number of observations available over the time period of interest. This may also

be evidence of noise in the SIT analysis, uncertainties in the modelled snow depth, in the assimilated SIT observations or in the data used for validation."

**Line 466-470:**

*"However, the model standard deviation and correlation coefficient are poorer on assimilation of these data. It should be noted that there are many more OIB observations over both the Canadian Arctic and Beaufort Sea regions (Fig. 12), and these agree much better with the model output and the CryoSat-2 observations than do the Air-EM observations. This potentially indicates an uncertainty in the quality of the Air-EM observations. Uncertainty in the snow depth will also be contributing to this issue, although the difference between the CryoSat-2 and Air-EM observations is greater than the snow depth itself."*

Changed to:

"However, the model standard deviation and correlation coefficient are poorer on assimilation of these data. It should be noted that there are many more OIB observations over both the Canadian Arctic and Beaufort Sea regions (compare Figs. 12 and 13; 547 matchups for OIB versus 45 for Air-EM), and these agree much better with the model output and the CryoSat-2 observations than do the Air-EM observations. This potentially indicates a sampling uncertainty in the Air-EM matchups. Uncertainty in the modelled snow depth will also be contributing to this issue, although the difference between the CryoSat-2 and Air-EM observations is greater than the snow depth itself."

**Line 522-524:**

*"The BGEP data for March-April 2015-2017 (Fig. 15(a)) compares better with the CryoSat-2 observations than does the Air-EM data in the Beaufort Sea for April 2015 (Fig. 13(a)). This again suggests that the Air-EM data may be unreliable, as discussed in Sect. 5.2."*

These lines have been removed.

**Line 543-548:**

*"Validation against springtime airborne electromagnetic induction (Air-EM) combined SIT and snow depth observations (Haas et al., 2009) yields poorer results than for the OIB and BGEP datasets, despite covering similar locations. This may be evidence of uncertainty in the Air-EM observations, or sampling error owing to the limited number of matchups available from this dataset. It may also be a result of noise in the SIT analysis, uncertainty in the modelled snow depth, or uncertainty in the assimilated observations."*

Changed to:

"Validation against springtime airborne electromagnetic induction (Air-EM) combined SIT and snow depth observations (Haas et al., 2009) yields poorer results than for the OIB and BGEP datasets, despite covering similar locations. This may be evidence of sampling uncertainty in the Air-EM matchups, owing to the more limited number of observations available from this dataset that cover the time period of interest. It may also be a result of noise in the SIT analysis, uncertainty in the modelled snow depth, in the assimilated observations, or in those used for validation."